# A DENSE SUBSET INDEX FOR COLLECTIVE QUERY COVERAGE

**Kartik Nair**[*2]**, Pritish Chakraborty**[1]**, Atharva Tambat**[1]**, Indradyumna Roy**[1]**,
Soumen Chakrabarti**[1]**, Anirban Dasgupta**[3]**, Abir De**[1]

[1]IIT Bombay, [2]Carnegie Mellon University, [3]IIT Gandhinagar
{pritish, atharvatambat, indraroy15,soumen,abir}@cse.iitb.ac.in
ksnair@cs.cmu.edu, anirbandg@iitgn.ac.in

## ABSTRACT

In traditional information retrieval (IR), corpus items compete with each other to occupy top ranks in response to a query. In contrast, in many recent retrieval scenarios such as multi-hop question answering (QA), table QA, or text-to-SQL, items are not self-complete: they must instead collaborate, i.e., information from multiple items must be combined to respond to the query. In the context of modern dense retrieval, this need translates into finding a small collection of corpus items whose contextual word vectors collectively cover the contextual word vectors of the query. The central challenge is to retrieve a near-optimal collection of covering items in time that is sublinear in corpus size. By establishing coverage as a submodular objective, we enable successive dense index probes to quickly assemble an item collection that achieves near-optimal coverage. Successive query vectors are iteratively 'edited' to account for current coverage, and the dense index is built using random projections of a novel, lifted dense vector space. Beyond rigorous theoretical guarantees, we report on a scalable implementation of this new form of vector database. Extensive experiments establish the empirical success of our method, in terms of the best coverage vs. query latency tradeoffs. Code can be found at https://github.com/structlearning/DISCo.

## 1 INTRODUCTION

In traditional information retrieval (IR), the relevance score of an item (passage or document), given a query, is computed independent of other items (Manning et al., 2008), and top-$K$ items presented. The underlying assumption is that a relevant item is self-complete, and can satisfy the information need by itself. Therefore, different items are *alternatives* or *competitors*. The basic premise that items are competitors is invalid for many modern retrieval applications. In a multi-hop question answering (QA) setting (Yang et al., 2018), given the query "Which computer scientist from Clamart was an editor of Algorithmica?" the following Wikipedia passages *collaboratively* provide the answer — a competitive comparison of their relevance scores against each other is pointless.

**Mohammad T. Hajiaghayi,** Computer scientist at the University of Maryland, was recently chosen to serve as editor-in-chief of Algorithmica

**Algorithmica:** Editor's foreword; Bernard Chazelle. This special issue of Algorithmica is devoted to computational geometry.

**Bernard Chazelle:** Bernard Chazelle is a French computer scientist. Chazelle was born in Clamart.

Crucially, there may be no single passage with both Clamart and Algorithmica in it. Thus, isolated ranking can mislead (first passage), and reasoning over *item subsets* is essential for full query coverage. Similar challenges arise in knowledge graph QA (Kosten et al., 2023), table QA (Chen et al., 2021; Zhao et al., 2022b; Chen et al., 2025b), and schema retrieval for text-to-SQL (Li et al., 2023; Lei et al., 2025; Chen et al., 2025a), where evidence spans multiple corpus items. In all these applications, a relevance score must be associated with an *item subset*, not individual items. Implementations often use off-the-shelf vector databases with per-item scoring functions with an inflated

---

[*]Work done while at IIT Bombay

$K$, hoping to recall a superset of the corpus items needed, then filter away the rest. This two-stage approach can fall short because of two related reasons: top-scoring items (e.g., multiple biographies of Chazelle) within the budget of $K$ may be redundant, and yet fail to cover all aspects of the query.

**Importance of collective query coverage** These limitations point to a central lesson: beyond relevance of *individual* corpus items, a retriever should promote item subsets that *collectively cover* the query. Our goal is to design and implement an efficient algorithm to achieve this end.

**Our contributions** Given the recent success of dense retrieval (Zhao et al., 2022a) and its use with language models (Lewis et al., 2021), a new **formulation** for *collective first-stage retrieval* is crucial, along with a **scalable algorithm** and an **implementation recipe** of this new form of a vector database. We introduce advances on all three fronts.

— *Collective retrieval as vector bag coverage:* We model the query and each corpus item as a subset of a universe of atoms (*e.g.*, words from a vocabulary), mapped into bags of dense embedding vectors via a contextual encoder (Reimers et al., 2019; Lee et al., 2019). Then, we compute the score of a subset $S$ of items by measuring the extent to which each atom vector $q \in Q$ is 'covered' by *some* atom vector in $\{X_c : c \in S\}$. This allows us to represent the collective retrieval problem as maximization of facility-location (Cornuejols et al., 1977) style *coverage objective*, which evaluates the extent of query coverage achieved by the selected subset. Our proposal generalizes Chamfer based MaxSim score (Butt et al., 1998; Khattab et al., 2020; Santhanam et al., 2021; 2022) and is monotone and submodular. As a result, a greedy algorithm—where each iteration adds the document with maximum marginal gain—enjoys an approximation guarantee, while avoiding exponential complexity.

— *Marginal gain maximization viewed as retrieval:* Direct evaluation of marginal gains over all documents is infeasible for large corpora, as it scales linearly with corpus size at each selection round. We view this problem as a classic retrieval challenge, albeit with a more complex objective: find the item with the highest relevance score, defined as the marginal gain.

— *ANN retrieval using marginal gain approximation:* Efficient retrieval requires fast indexed access of the corpus items. However, the marginal gain is not directly amenable to indexing the corpus items in a manner that allows sublinear-time access to items with the largest marginal utility. To get past this blocker, we first propose a novel 'lifted' or augmented vector representation of query and corpus atoms. Specifically, we encode the evolving coverage of the current set $S$ at each iteration into an *augmented query representation*, while corpus embeddings remain static. This expresses the marginal gain as a sum of hinge functions over dot products of the 'lifted' representations, which, however, still remains difficult to index. Then we design a novel random projection of the lifted representations, yielding a form that is finally amenable to indexing. We give an end-to-end probabilistic approximation guarantee of the whole method, which we call DISCO (**D**ense **I**ndex for **S**et **Co**verage). (Here "dense" means the embedding vectors are dense and not sparse, e.g., in discrete token space.)

— *Implementation and experiments:* We implement DISCO as a multi-vector dense retrieval architecture optimized for I/O and computational efficiency, and balancing query latency against subset selection quality. Experiments across standard benchmarks demonstrate that DISCO achieves superior query coverage at a given rank, with significantly better latency–quality tradeoffs—often exceeding $100\times$ speedups over greedy baselines.

## 2 PRELIMINARIES

**Notation** We define $[N] = \{1, \ldots, N\}$, and $[\cdot]_+ = \max\{\cdot, 0\}$ as the ReLU or hinge function. Given a vocabulary of atoms (which can be words, tokens, or other elementary components) $\mathcal{V}$, A query is modeled as a set or sequence of atoms $(q_1, \ldots, q_M) \subset \mathcal{V}^M$. Similarly, we write a corpus item as $(x_1, \ldots, x_L) \subset \mathcal{V}^L$. We write $q_\bullet \in \mathbb{R}^d$ and $x_\bullet \in \mathbb{R}^d$ as the contextual dense embeddings vectors of $q_\bullet$ and $x_\bullet$. If queries and items are sets of atoms, we can convert them to sets of vectors, one per atom, using a set transformer (Lee et al., 2019). If queries and items are sequences, a transformer-based language model (Reimers et al., 2019) can convert them to sets of vectors, one per atom. Contextualization lets us represent the query as a **set** of dense embeddings $Q = \{q_1, \ldots, q_M\}$, and similarly the corpus item as $X = \{x_1, \ldots, x_L\}$. Given the indices of the corpus items $\mathcal{C} = [N]$, we write $X_c$ to denote one corpus item, indexed with $c \in \mathcal{C}$. Throughout, we assume $\|q\| = \|x\| = 1$ for all $q \in Q$ and $x \in X$.

**Monotonicity and submodularity** Given the set of corpus items $\{X_c \mid c \in \mathcal{C}\}$, we consider a query dependent set function $F(\bullet, Q) : 2^{\mathcal{C}} \to \mathbb{R}$. For a subset $S \subset \mathcal{C}$, we denote the **marginal gain** of $c$ beyond $S$ as $F(c \mid S, Q) = F(S \cup \{c\}, Q) - F(S, Q)$. Given a query $Q$, the function $F(\bullet, Q)$ is called monotone if $F(c \mid S, Q) \geq 0$ whenever $S \subset \mathcal{C}$ and $c \in \mathcal{C} \backslash S$; and, $F$ is called submodular if $F(c \mid S, Q) \geq F(c \mid T, Q)$ for $S, T \subset \mathcal{C}$ and $c \in \mathcal{C} \backslash T$ (Edmonds, 1970).

**MaxSim (or Chamfer) score** (Khattab et al., 2020; Santhanam et al., 2021; 2022; Dhulipala et al., 2024) $\mathrm{MaxSim}(Q, X)$ is a relevance score between a query $Q$ and a single document $X$. Informally, it measures the extent to which a corpus item $X$ 'covers' the atoms/words of a query $Q$. For each atom vector $\boldsymbol{q} \in Q$, $\mathrm{MaxSim}$ identifies the most similar atom vector in $X$, then aggregates these maximum similarities across all query atoms. Formally,

$$\mathrm{MaxSim}(Q, X) = \sum_{\boldsymbol{q} \in Q} \max_{\boldsymbol{x} \in X} \boldsymbol{q}^{\top} \boldsymbol{x} \tag{1}$$

Note that multiple query atoms may be covered by a single atom in a corpus item. The distance analogue of this similarity is the well-known *Chamfer distance* (Borgefors, 1988; Butt et al., 1998; Ma et al., 2010; Feng et al., 2025).

**Independent top-$K$ retrieval** Current late-interaction retrievers, such as ColBERT (Khattab et al., 2020) and its variants (Santhanam et al., 2021; 2022), rank documents independently using the $\mathrm{MaxSim}$ score (1). Retrieving the top-$K$ items under this scoring is equivalent to solving the optimization problem: $\max_{S \subset \mathcal{C}:|S|=K} \sum_{c \in S} \mathrm{MaxSim}(Q, X_c)$, which simply selects the $K$ documents with the largest individual $\mathrm{MaxSim}$ scores. In other words, this optimization reduces to independently scoring all documents and returning the $K$ highest scorers, without accounting for redundancy or interaction among the selected items.

To support efficient search for top-$K$ independent items, ColBERT and its variants design indexing and pruning techniques tailored to $\mathrm{MaxSim}$. In particular, they employ an inverted file index (IVF) (Douze et al., 2024), where atom/word vectors $\boldsymbol{x} \in X_c$ from all corpus item are clustered into centroids. At query time, for each query atom/word vector $\boldsymbol{q} \in Q$, the index retrieves candidate corpus items associated with nearby centroids, followed by successive filtering and pruning stages that approximate the final $\mathrm{MaxSim}$ score while maintaining efficiency.

**Limitations of top-$K$ retrieval for complex queries** The top-$K$ retrieval problem, discussed above, seeks $K$ corpus items from $\mathcal{C}$ such that *each item individually* covers the query $Q$, independent of any other item in the top-$K$ list. However, for a class of complex queries, this approach is suboptimal, as can be seen from our running example. No single passage may cover all atoms/words in the query, but multiple passages may cover most query atoms/words and contain enough information to answer the question.

**Overview of our approach** To tackle the above challenges, we avoid retrieving top-$K$ independent items based on MaxSim score (1). Instead, we design a relevance measure based on soft set coverage, which guides the selection of a subset $S \subset \mathcal{C}$, which can collectively cover the query $Q$, instead of each covering $Q$ independently. Subsequently, we design an indexing and search mechanism, specifically tailored to maximizing this coverage objective.

## 3 PROPOSED APPROACH

We design a coverage-based set utility function $F(S, Q)$, guided by the related Facility Location objective (Cornuejols et al., 1977; Mirchandani et al., 1990; Lin et al., 2009). We first present our objective in the context of retrieval and describe the well-known greedy method (Nemhauser et al., 1978), which can maximize $F$ by iteratively searching over the entire corpus set $\{X_c\}$. Next, we provide an approximation of the marginal gain $F(c \mid S, Q)$, which is amenable for indexing and search. Finally, we describe our multivector inverted file index (IVF) and the associated query-time search procedure, both tailored to the coverage-based objective.

### 3.1 COVERAGE MAXIMIZATION

**Objective** Let $\{X_c \mid c \in \mathcal{C}\}$ be a large collection of corpus items. We seek to maximize a coverage objective function $F(S, Q)$, which quantifies how well the set $S$ collectively covers the atom embeddings $\boldsymbol{q} \in Q$. Given a query $Q$, for each atom $\boldsymbol{q} \in Q$, we first compute the maximum similarity $q$ attains with any atom across the corpus items in $S$ and then aggregate over all $\boldsymbol{q} \in Q$. Formally,

we write our set utility $F(S, Q)$ and coverage-based retrieval objective as:

$$F(S, Q) = \sum_{\boldsymbol{q} \in Q} \max_{\boldsymbol{x} \in \cup_{c \in S} X_c} \boldsymbol{q}^\top \boldsymbol{x}; \qquad \underset{S \subset \mathcal{C}}{\text{maximize}} \; F(S, Q) \text{ such that } |S| \leq K. \qquad (2)$$

In the optimization for top-$K$ retrieval task, *viz.*, $\max_{S \subset \mathcal{C}:|S|=K} \sum_{c \in S} \text{MaxSim}(Q, X_c)$, it is the outer sum, independently carried over $c \in S$, which can introduce redundancy, potentially returning multiple documents that cover the same query atom/s. In contrast, $F(S, Q)$ (2) does not give additive credit for multiple document items redundantly covering the same query atom.

**Monotonicity, submodularity and greedy maximization**  From the extensive literature on submodular functions (Cornuejols et al., 1977; Lin et al., 2009; Krause et al., 2014), it is easy to establish that $F(S, Q)$ is monotone and submodular. Therefore, we can obtain an approximate solution of the maximization prob-
lem (2), using a greedy algorithm (Nemhauser et al., 1978). Having computed a set $S_k$ at iteration $k \leq K$, this algorithm iteratively selects the item having index $c$ that has the highest marginal gain $F(c \mid S, Q)$ at each step (Algorithm 1). While not guaranteed to find the true optimal set $S^* = \arg\max_{S \subset \mathcal{C}:|S| \leq K} F(S, Q)$, it is guaranteed to find a solution that is within a constant factor of $(1 - 1/e) \approx 63\%$ of the optimal solution. In practice, it is frequently found much closer to optimal.

---
**Algorithm 1** Greedy algorithm to solve (2).

---
1: Initialize $S_0 \leftarrow \emptyset$
2: **for** $k = 1, \ldots, K$ **do**
3:     $c_k = \arg\max_{c \in \mathcal{C} \setminus S_{k-1}} F(c \mid S_{k-1}, Q)$
4:     $S_k \leftarrow S_{k-1} \cup \{c_k\}$
5: **return** $S_K$

---

**Bottlenecks with greedy algorithm**  Despite its theoretical appeal, the greedy algorithm is prohibitively expensive for large corpora, as it requires $\Theta(|\mathcal{C}|)$ evaluations of the marginal gain $F(X \mid S_{k-1}, Q)$ per iteration (line 3 of Algorithm 1). For the MS-MARCO corpus ($|\mathcal{C}| \sim 8.8$ million passages), selecting just $K = 10$ corpus items requires 88 million marginal gain computations. Even at a speed of one millisecond per computation, this retrieval process would take over 24 hours for a single query, which is prohibitive.

## 3.2  BRIEF DISCUSSION OF RELATED WORK

**Existing variants of greedy selection**  For atomic (i.e., non-embedded) items, the scalability of set cover has been extensively studied (Cormode et al., 2010). Existing alternatives such as Lazy Greedy (Minoux, 2005), Stochastic Greedy (Mirzasoleiman et al., 2015), and Lazier-than-lazy Greedy (Mirzasoleiman et al., 2015) offer partial efficiency gains. Lazy Greedy reduces the number of evaluations by maintaining a heap but still relies on exhaustive scoring. Stochastic Greedy and Lazier-than-lazy Greedy prune the corpus uniformly at random in each iteration, in a query-agnostic manner that degrades utility. As a result, these works do not provide a desirable trade off between efficiency and utility.

**Diversity in information retrieval**  Submodular set reward functions have been proposed in the Information Retrieval community since at least 1998, motivated by diversity (Bennett et al., 2008) and subtopic coverage (Zhai et al., 2003). These objectives are usually implemented as a reranking stage, after the small subset of candidates has already been selected using a scalable first-stage retriever. For reranking, max marginal relevance (Carbonell et al., 1998), multi-armed bandits (Radlinski et al., 2008), determinantal point processes (Kulesza, 2012a; Chen et al., 2017), query reformulation (Santos et al., 2010), etc., are used. Hence, these approaches focus on scoring function computation at the reranking stage. These reranking efforts are vulnerable to loss of recall in the first-stage. We contrast these approaches with ours, and provide a further discussion of related work in Appendix E.

## 3.3  RETRIEVAL-ORIENTED APPROXIMATION OF MARGINAL GAIN

**Marginal gain maximization from the viewpoint of ANN retrieval**  To address the above bottlenecks, we need a method to *efficiently* retrieve the candidates with high marginal gain, in a query dependent manner. In line with traditional IR, we view the task of maximizing the marginal gain as "retrieving the corpus item $c$ with the largest relevance score $F(c \mid S_{k-1}, Q)$" at each iteration $k$. This requires us to design a retrieval model tailored to the relevance score $F(c \mid S_{k-1}, Q)$, along with compatible indexing and search techniques. However, there are two key challenges:

**(1)** Unlike dot-product or cosine similarity, the marginal gain $F(c \mid S_{k-1}, Q)$ bears a complex relationship between $Q$ and $X_c$, which is not readily supported by standard indexing models.

**(2)** For each iteration $k$, the subset $S_k$ depends heavily on the query $Q$. This poses a challenge for indexing, which should ideally employ query-agnostic, one-time preprocessing.

To this end, we approximate the marginal gain $F(c \,|\, S, Q)$, to make it amenable to design an approximate nearest neighbor (ANN) retrieval model. We perform this approximation in two steps: (I) We provide an alternative representation of $F(c \,|\, S, Q)$ using the dot products of two augmented vectors, one depending on the query and other on corpus items. (II) We design a method, based on random projections, to build a scalable ANN data structure.

**Marginal gain computation using vector augmentation**  We strategically rewrite the marginal gain $F(c \,|\, S, Q)$ as follows:[1]

**Proposition 1.** *Given the coverage objective $F(S, Q)$ defined in Eq. (2), the marginal gain is:*

$$F(c \,|\, S, Q) = \sum_{\boldsymbol{q} \in Q} \max_{\boldsymbol{x} \in X_c} \left[ \boldsymbol{q}^\top \boldsymbol{x} - \max_{u \in S} \max_{\boldsymbol{x}' \in X_u} \boldsymbol{q}^\top \boldsymbol{x}' \right]_+ \tag{3}$$

The above formula reveals that the total marginal gain of a new corpus item $c$ is the sum, over all query atom embeddings $\boldsymbol{q} \in Q$, of the *new* score contribution $(\boldsymbol{q}^\top \boldsymbol{x})$ that exceeds the best score already achieved for that atom by the current set $S$.

Designing indexing and search techniques for any relevance measure requires expressing the score as a similarity between precomputable corpus representations and a query representation that is corpus-agnostic. However, in the second term inside the hinge of Eq. (3), the corpus token embeddings $\boldsymbol{x}'$ are coupled with $S$, which itself depends on the query $Q$ through previously selected items. To isolate $\boldsymbol{x}'$ from $S$ and $Q$, we reformulate this term as follows. We observe that $\max_{u \in S} \max_{\boldsymbol{x}' \in X_u} \boldsymbol{q}^\top \boldsymbol{x}' = F(S, \boldsymbol{q})$ denotes the set coverage value for a singleton set with atom $q$ alone. If we define the augmented representations of the query and corpus tokens as:

$$\widehat{\boldsymbol{q}}_S := [\boldsymbol{q}; F(S, \boldsymbol{q})] \in \mathbb{R}^{d+1}, \quad \widehat{\boldsymbol{x}} := [\boldsymbol{x}; -1] \in \mathbb{R}^{d+1}, \tag{4}$$

then we express Eq. (3) as a dot product between these two augmented vectors in lifted vector space:

$$F(c \,|\, S, Q) = \sum_{\boldsymbol{q} \in Q} \max_{\boldsymbol{x} \in X_c} [\boldsymbol{q}^\top \boldsymbol{x} - F(S, \boldsymbol{q})]_+ = \sum_{\boldsymbol{q} \in Q} \max_{\boldsymbol{x} \in X_c} [\widehat{\boldsymbol{q}}_S^\top \widehat{\boldsymbol{x}}]_+ \tag{5}$$

Here, we transfer the dependency of $F$ on $S$ into the query and obtain a state-dependent representation $\widehat{\boldsymbol{q}}_S$. Consequently, the corpus token representation $\widehat{\boldsymbol{x}}_S$ becomes agnostic to $S$ and $Q$.

**Random projection for hinge-ANN**  The last hurdle we face is the non-linearity of the hinge function, $[\bullet]_+$ in Eq. (5), which prevents direct use of a standard ANN retrieval. To address this challenge, we approximate $[\widehat{\boldsymbol{q}}_S^\top \widehat{\boldsymbol{x}}]_+$ using dot product of two vectors, obtained by projecting $\widehat{\boldsymbol{q}}_S$ and $\widehat{\boldsymbol{x}}$ on to random hyperplanes. The key insight is to use a randomized feature map that, with high probability, mimics the behavior of the hinge function applied to dot product between two vectors.

Let $\boldsymbol{u}, \boldsymbol{v} \in \mathbb{R}^{d+1}$ be two vectors (e.g., $\boldsymbol{u} = \widehat{\boldsymbol{q}}_S$ and $\boldsymbol{v} = \widehat{\boldsymbol{x}}$); and $\mathbf{w} \in \mathbb{R}^{d+1}$ be a random vector drawn from a standard multivariate normal distribution, i.e., $\mathbf{w} \sim \mathcal{N}(0, \boldsymbol{I}_{d+1})$. We define a feature map $\Phi_{\mathbf{w}} : \mathbb{R}^{d+1} \to \mathbb{R}^{2(d+1)}$ as follows:

$$\Phi_{\mathbf{w}}(\boldsymbol{u}) \triangleq \left[ \boldsymbol{u}; \; \mathrm{sign}(\mathbf{w}^\top \boldsymbol{u}) \cdot \boldsymbol{u} \right] / \sqrt{2}. \tag{6}$$

Then, we can express $[\boldsymbol{u}^\top \boldsymbol{v}]_+$ as follows.

**Theorem 2.** *Given any two vectors $\boldsymbol{u}, \boldsymbol{v} \in \mathbb{R}^{d+1}$, let $\mathbf{w}$ be a random hyperplane $\mathbf{w} \sim \mathcal{N}(0, \boldsymbol{I}_{d+1})$, and $\Phi_{\mathbf{w}}$ be the transformation defined in Eq. (6). Then, we have the following result:*

$$\Phi_{\mathbf{w}}(\boldsymbol{u})^\top \Phi_{\mathbf{w}}(\boldsymbol{v}) = [\boldsymbol{u}^\top \boldsymbol{v}]_+ \qquad \text{with probability } p \geq 0.5. \tag{7}$$

**Proof sketch** Note that $\Phi_{\mathbf{w}}(\boldsymbol{u})^\top \Phi_{\mathbf{w}}(\boldsymbol{v}) = \boldsymbol{u}^\top \boldsymbol{v}$ or 0, based on whether $\mathrm{sign}(\mathbf{w}^\top \boldsymbol{u}) = \mathrm{sign}(\mathbf{w}^\top \boldsymbol{v})$ or not. From Charikar (2002), we can show that probability of this condition is $p = 1 - \frac{1}{\pi} \arccos(\frac{\boldsymbol{u}^\top \boldsymbol{v}}{||\boldsymbol{u}|| \cdot ||\boldsymbol{v}||})$ or $p = \frac{1}{\pi} \arccos(\frac{\boldsymbol{u}^\top \boldsymbol{v}}{||\boldsymbol{u}|| \cdot ||\boldsymbol{v}||})$. We use the sign of $\boldsymbol{u}^\top \boldsymbol{v}$ to argue that $p \geq 0.5$.

While the above relationship holds with $p \geq 1/2$, we can draw more random hyperplanes $\mathbf{w}_r$ and perform a max aggregation of $\Phi_{\mathbf{w}_r}(\boldsymbol{u})^\top \Phi_{\mathbf{w}_r}(\boldsymbol{v})$ to obtain a more accurate estimate of $[\boldsymbol{u}^\top \boldsymbol{v}]_+$. This gives us our approximation result as follows.

**Theorem 3.** *Suppose $\{\mathbf{w}_1, \dots, \mathbf{w}_R\}$ be $R$ random hyperplanes with $\mathbf{w}_r \overset{i.i.d.}{\sim} \mathcal{N}(0, \boldsymbol{I}_{d+1})$. Given the state-dependent representation for each query token $\widehat{\boldsymbol{q}}_S := [\boldsymbol{q}; F(S, \boldsymbol{q})]$ and the query agnostic*

---

[1]Proofs of all technical results are in Appendix F.

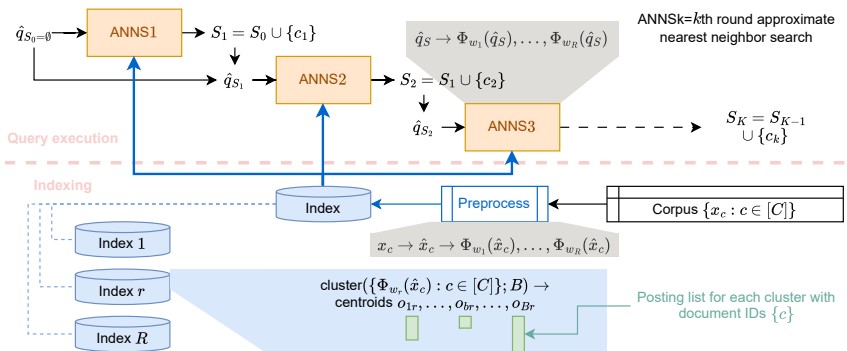

Figure 1: DISCo block diagram. Random vectors $\{\boldsymbol{w}_r : r \in [R]\}$ are sampled. Corpus items $\{x_c : c \in [C]\}$ are mapped to vectors $\Phi_{\boldsymbol{w}_r}(\hat{\boldsymbol{x}}_c)$ for each $r$. A sub-index is built for each $r$. In it, we cluster $\{\Phi_{\boldsymbol{w}_r}(\hat{\boldsymbol{x}}_c) : c \in [C]\}$ into $B$ clusters with centroids $\{o_{br} : b \in [B]\}$. Each centroid is associated with a posting list containing document IDs $\{c\}$. Query execution proceeds in $K$ rounds, while the retrieved item set grows from $S_0 = \varnothing$ to $S_K$ as output. Each round executes an approximate nearest neighbor search, labeled ANNS1 to ANNS$K$. Each ANNS$k$ round consults all $R$ indices and chooses the next item $c_k$ in $o(C)$ time.

*representation of each corpus token $\widehat{\boldsymbol{x}} := [\boldsymbol{x}; -1]$ as defined in Eq. (4), let the randomized function $G_{\mathbf{w}_{1:R}}$ be defined as:*

$$G_{\mathbf{w}_{1:R}}(c; S, Q) = \sum_{\boldsymbol{q} \in Q} \max_{r \in [R]} \max_{\boldsymbol{x} \in X_c} \Phi_{\mathbf{w}_r}(\widehat{\boldsymbol{q}}_S)^\top \Phi_{\mathbf{w}_r}(\widehat{\boldsymbol{x}}) \tag{8}$$

*Then, for any $c \notin S$, $S \subset \mathcal{C}$ and $Q = \{q\}$, a positive probability $\delta > 0$ and $R \geq \log(|Q|/\delta)$, we will have $G_{\mathbf{w}_{1:R}}(c; S, Q) = F(c \,|\, Q, S)$, with probability atleast $1 - \delta$.*

**Proof sketch** We note that $\max_{\mathbf{w}_r \in \mathbb{R}^{d+1}} \Phi_{\mathbf{w}_r}(\widehat{\boldsymbol{q}}_S)^\top \Phi_{\mathbf{w}_r}(\widehat{\boldsymbol{x}}) = [\widehat{\boldsymbol{q}}_S^\top \widehat{\boldsymbol{x}}]_+$. Therefore, we draw multiple samples of $\mathbf{w}_r$ and compute an empirical maxima of the dot product between the projected vectors which approximate $[\widehat{\boldsymbol{q}}_S^\top \widehat{\boldsymbol{x}}]_+$. Then obtain the required probability using union bound.

We observe that if the number of tokens $|Q| < 32$ — which is a common scenario in applications — then we need only $R = 8$ hyperplanes to ensure that $G_{\mathbf{w}_{1:R}}(c; S, Q) = F(c \,|\, Q, S)$ with probability at least $1 - \delta = 0.875$. If the query were shorter, say, $|Q| < 16$, the success probability would increase to $1 - \delta = 0.94$. (The union bound is likely pessimistic in practice.) Note that

---

**Algorithm 2** Greedy algorithm based on proposed approximation

1: Initialize $S_0 \leftarrow \emptyset$, $\mathbf{w}_1, ...., \mathbf{w}_R \sim \mathcal{N}(0, \boldsymbol{I}_{d+1})$
2: **for** $k = 1, \ldots, K$ **do**
3:    $c_k = \arg\max_{c \in \mathcal{C} \setminus S_{k-1}} G_{\mathbf{w}_{1:R}}(c; S, Q)$
4:    $S_k \leftarrow S_{k-1} \cup \{c_k\}$
5: **return** $S_K$

---

the RHS of Eq. (8) closely resembles the MaxSim score in Eq. (1), which is amenable to indexing and search. In the next section, we develop indexing and search methods tailored to $G_{\mathbf{w}_{1:R}}$.

**Approximation guarantee** In Algorithm 2, we replace the marginal gain in line 3 of Algorithm 1 with our approximation $G_{\mathbf{w}_{1:R}}(c; S_{k-1}, Q)$. In line 3 of Algorithm 2, we obtain an element $c$ such that $F(c \,|\, S_{k-1}, Q)$ with probability at least $1 - \delta$. This algorithm can be shown to enjoy an optimality guarantee (in expectation) of $(1 - 1/e - \delta)$, which is very close to Algorithm 1, even with small value of $R = 8$ as discussed above.

**Theorem 4** (Approximation guarantee of Algorithm 2). *Let $S_K$ be the set of $K$ corpus items selected by Algorithm 2, and let $S^*$ be the optimal set for $F$, i.e., $S^* \in \arg\max_{S:|S| \leq K} F(S, Q)$. Given $\delta \in (0, 1]$, if we set the number of random projection vectors $R \geq \log(|Q|/\delta)$,*

$$\mathbb{E}_{\mathbf{w}_{1:R}}[F(S_K, Q)] \geq (1 - 1/e - \delta) \cdot F(S^*, Q). \tag{9}$$

### 3.4 INDEXING AND RETRIEVAL

We complete the presentation of our system, DISCo, by designing an ANN index to implement the search for $\arg\max_c G_{\mathbf{w}_{1:R}}(c, S_{k-1}, Q)$ in every round, building upon the best practices from the ColBERT family of multi-vector ANN methods: representing each item as a multivector (a set of token embeddings) (Khattab et al., 2020), storing them compactly via centroid–residual encoding (Santhanam et al., 2021), and accelerating candidate filtering through centroid-first prun-

ing (Santhanam et al., 2022). However, significant modifications and enhancements are needed to support our collective objective, lifted vector space, mutating queries and multiple projections. A block diagram of DISCo is shown in Figure 1.

**Indexing** Our indexing step is based on multivector IVF, adapted from Santhanam et al. (2022, PLAID), but tailored to maximizing the relevance score measured in terms of $G_{\mathbf{w}_{1:R}}(\bullet, S_{k-1}, Q)$. Algorithm 3 summarizes the DISCo indexer. Corpus atom vectors are augmented in step 4, which will support query-dependent, per-round coverage subtraction ($-F(S, \boldsymbol{q})$ in Eq. (5)), while remaining query-agnostic. $R$ projection vectors are sampled in step 6. For each projection vector, feature maps $\Phi_{\mathbf{w}_r}(\widehat{\boldsymbol{x}})$ are computed in step 7. To prepare the IVF index, the feature maps are clustered to obtain a set of centroids in step 9. Each $\Phi_{\mathbf{w}_r}(\widehat{\boldsymbol{x}})$ is then represented by its nearest centroid $\boldsymbol{o}$ and a compact residual code $\boldsymbol{\Delta}$ produced by a Quantizer (Santhanam et al., 2021; 2022) to reduce the number of bits needed per dimension. For each sample index $r \in [R]$, we return (1) the projection vector $\mathbf{w}_r$, (2) cluster centroids $\{\boldsymbol{o}_{b,r}\}_{b=1}^{B}$, (3) inverted lists $\mathrm{InvInd}^{(r)}$, and (4) forward index $\mathrm{FwdInd}^{(r)}$, mapping every document $c$ to its sequence of $(b^*, \boldsymbol{\Delta})$ codes across atoms.

---

**Algorithm 3** DISCo indexing stage.

1: **Input:** Corpus embeddings $\{X_c = \{\boldsymbol{x} \in \mathbb{R}^d\}\}_{c \in \mathcal{C}}$; #hyperplanes $R$; #clusters $B$; Quantizer (see text)
2: **Output:** Inverted index $\mathrm{InvInd}^{(r)}$; forward index $\mathrm{FwdInd}^{(r)}$, centroids $\{\boldsymbol{o}_{b,r}\}_{b=1}^{B}$ for $r \in [R]$
3: $\mathrm{InvInd}^{(r)}, \mathrm{FwdInd}^{(r)} \leftarrow \varnothing\ \forall r$
4: **Augment:** $\widehat{\boldsymbol{x}} \leftarrow [\boldsymbol{x}; -1]$ for all $\boldsymbol{x} \in X_c, c \in \mathcal{C}$ (4)
5: **for** $r = 1$ **to** $R$ **do**
6:     **Random sample:** $\mathbf{w}_r \sim \mathcal{N}(0, \boldsymbol{I}_{d+1})$
7:     **Project and prepare postings:**
8:     $\mathcal{Z} \leftarrow \{(\Phi_{\mathbf{w}_r}(\widehat{\boldsymbol{x}}), c) \mid \widehat{\boldsymbol{x}} \in X_c, c \in \mathcal{C}\}$ (6)
9:     **Cluster:** $\{\boldsymbol{o}_{b,r}\}_{b=1}^{B} \leftarrow \mathrm{kmeans}(\{\Phi_{\mathbf{w}_r}(\widehat{\boldsymbol{x}})\}, B)$
10:     **for** each $(\boldsymbol{z}, c) \in \mathcal{Z}$ **do**
11:         $b^* \leftarrow \mathrm{argmin}_b \|\boldsymbol{z} - \boldsymbol{o}_{b,r}\|_2$ {nearest centroid}
12:         $\boldsymbol{\Delta} \leftarrow \mathrm{Quantizer}(\boldsymbol{z} - \boldsymbol{o}_{b^*,r})$
13:         **Insert into indices:**
14:         $\mathrm{InvInd}^{(r)}(b) \leftarrow \mathrm{InvInd}^{(r)}(b^*) \cup \{c\}$
15:         $\mathrm{FwdInd}^{(r)}(c) \leftarrow \mathrm{FwdInd}^{(r)}(c) \cup (b^*, \boldsymbol{\Delta})$
16: **return** $\{(\mathbf{w}_r, \{\boldsymbol{o}_{b,r}\}_{b=1}^{B}, \mathrm{InvInd}^{(r)}, \mathrm{FwdInd}^{(r)})\}_{r=1}^{R}$

---

**Retrieval** Algorithm 4 shows how the index prepared thus far is used to respond to a query. Query processing proceeds in $K$ greedy rounds. In each round, each replica $r \in [R]$ contributes candidate items (documents). These are merged and pruned. We progressively refine candidates through six pruning stages, employing graded approximations to Eq. (8) and ultimately converging toward the "gold standard" Eq. (5). Surviving candidates get their scores more accurately calculated by accessing their residuals and the cluster centroids. A final reranking is performed using the 'true' scores to select the candidate with the best marginal score for the current round. Then the next round commences. We elaborate on some salient steps of Algorithm 4 below.

— *Replica-level coarse filtering:* For each round $k$ and each projection $r$, for each query atom vector $\boldsymbol{q}$, we compute $\widehat{\boldsymbol{q}}_S$, then the vector $\Phi_{\mathbf{w}_r}(\widehat{\boldsymbol{q}}_S)$, and use it to probe $\mathrm{InvInd}^{(r)}$ to collect some corpus items. We compute the union of these item candidate sets over all query atoms, providing the roughest approximation to Eq. (8).

— *Replica-level centroid pruning:* To refine the initial candidate pools $\mathcal{C}_{r,0}$, we assign coarse relevance scores using query–centroid similarities. For each document item $c \in \mathcal{C}_{r,0}$, the forward index $\mathrm{FwdInd}^{(r)}$ provides the atom centroids associated with $c$, and the score of $c$ is re-estimated as shows in step 11. This step moves us one step closer to Eq. (8) by aggregating across query tokens, while still omitting the $\max_{r \in [R]}$ over replicas and utilizing centroids rather than the true corpus token embeddings. Following PLAID (Santhanam et al., 2022), centroids with scores below $\tau$ are pruned. Each replica thus retains a refined set $\mathcal{C}_{r,1}$ of $n$ candidates.

— *Replica Pooling:* We collect all the corpus items in each refined set for each replica into $\mathcal{C}_1$, *i.e.*, $\mathcal{C}_1 \leftarrow \bigcup_{r=1}^{R} \mathcal{C}_{r,1}$.

— *Fine-grained filtering:* The score of each corpus item $c \in \mathcal{C}_1$ is refined to

$$\sum_{\boldsymbol{q} \in Q} \max_{r \in [R]} \max_{\boldsymbol{o} \in \mathrm{FwdInd}^{(r)}(c)} \Phi_{\mathbf{w}_r}(\widehat{\boldsymbol{q}}_S)^{\top} \boldsymbol{o},$$

which pools information from all replicas and approximates Eq. (8) better, while still being centroid-based. The pool is then pruned to retain roughly $n/4$ candidates, denoted $\mathcal{C}_2$, for the final stage.

— *Residual scoring:* At this stage, we access residuals $\boldsymbol{\Delta}$ from forward indices $\mathrm{FwdInd}$ to reconstruct document items and score them more accurately. For each candidate $c \in \mathcal{C}_2$, we obtain a set of reconstructed token embeddings $T_c = \{\widehat{\boldsymbol{x}}_{\boldsymbol{\Delta}} = \boldsymbol{\Delta} + \boldsymbol{o}\}$ where $(\boldsymbol{o}, \boldsymbol{\Delta}) \in \mathrm{FwdInd}^{(r)}(c)$, serving as an approximation to the original $X_c$.

The marginal gain of $c$ is refined as $\sum_{\boldsymbol{q} \in Q} \max_{r \in [R]} \max_{\hat{\boldsymbol{x}} \in T_c} \Phi_{\mathbf{w}_r}(\hat{\boldsymbol{q}}_S)^\top \hat{\boldsymbol{x}}$, which closely mirrors Eq. (8) while still operating on compressed rather than full-precision embeddings. The candidates are reranked accordingly, and the top $n'$ items retained in $\mathcal{C}_3$.

— *Full-precision scoring+selection:* $\mathcal{C}_3$ is small enough that we can now compute the exact marginal gain to the coverage objective $F(c \mid S, Q)$ (5) for each surviving $c \in \mathcal{C}_3$. The gain is evaluated using the full atom vectors in $X_c$, without centroid or residual approximations. Item $c_k$ with the largest gain is accumulated to form $S_k$ from $S_{k-1}$. The query representation is updated as $\hat{\boldsymbol{q}}_S \leftarrow \hat{\boldsymbol{q}}_{S \cup \{c_k\}}$, in preparation for the next round.

## 4  EXPERIMENTS

We provide a comprehensive evaluation of DISCO using seven real datasets and show that DISCO trades off between accuracy and efficiency better than several baselines, with striking improvements in efficiency. For details, see Appendices H and I.

**Datasets** We perform experiments with seven large-scale datasets, *viz.*, (1) MS-Marco, (2) HotpotQA and (3) Fever, from the BEIR (Thakur et al., 2021) benchmark; and, (4) Pooled, (5) Technology, (6) Writing and (7) Science from the LoTTE (Santhanam et al., 2021) benchmark. Among them, we report results on the first four datasets in the main and rest in Appendix H. Appendix G contains details about these datasets. Since Exact Greedy is prohibitively expensive, we use NFCorpus, a relatively smaller dataset from BEIR, to empirically evaluate the correctness of our theoretical results in one experiment.

**Baselines** We compare DISCO against seven competitive baselines, *viz.*, (1) Exact Greedy (Nemhauser et al., 1978), (2) Lazy Greedy (Minoux, 2005), (3) Stochastic Greedy (Mirzasoleiman et al., 2015), (4) Lazier-than-lazy Greedy (Mirzasoleiman et al., 2015), (5) PLAID (Santhanam et al., 2022), (6) MUVERA (Dhulipala et al., 2024) and (7) WARP (Scheerer et al., 2025) Among these, the first four methods target monotone submodular maximization, while the last three—collected from the information retrieval (IR) domain—focus on indexing followed by independent top-$K$ retrieval based on the MaxSim score in Eq. (1).

**Evaluation setting, coverage and efficiency** Each dataset consists of a query set $\mathcal{Q}$ and corpus items $\mathcal{C}$. We compute the contextual word embeddings $\boldsymbol{q}$ and $\boldsymbol{x}$ for query and corpus atoms/tokens, using bert-base-uncased (Devlin et al., 2018).

Our primary evaluation focus is the attained coverage $F(S_K, Q)$, where $S_K$ is the retrieved subset of size $K$. As motivation, fast ramp-up of coverage at small ranks is critical to economize energy on LLM-based downstream reasoners and response generators. We measure **coverage** averaged over queries, $\overline{F}_K = \sum_{Q \in \mathcal{Q}} F(S_K, Q)/|\mathcal{Q}|$. Moreover, given a method $\mathcal{M}$ and $K$, we measure the **efficiency** of $\mathcal{M}$ as $t_{\text{Exact Greedy},10}/t_{\mathcal{M},K}$, where $t_{\text{Exact Greedy},10}$ is the per-query time for Exact Greedy to retrieve $S_{10}$, and $t$ is the per-query retrieval time of $\mathcal{M}$ to retrieve $K$ items. We expect a inverse relationship between coverage and efficiency, and wish to identify methods with the best tradeoff.

**Traditional MAP, MRR, NDCG** In some datasets such as HotpotQA, test gold sets with $|S_{\text{gold}}| = 2$ are identified. All these items are needed to infer the response. In such cases, we can directly study the ranks at which these gold items appear. To condense these ranks to a single number, we can look

---

**Algorithm 4** DISCO query processing steps.

1: **Inputs:** Query $Q$, projections $\mathbf{w}_{1:R}$, centroids $\{\boldsymbol{o}_{b,r}\}$, indices $\{(\text{InvInd}^{(r)}, \text{FwdInd}^{(r)})\}$ for $b \in B, r \in [R]$
2: **Hyperparameters** $\tau, n, n'$
3: **Output:** Selected subset $S \subseteq \mathcal{C}$ of size $K$
4: Initialize $S_0 \leftarrow \emptyset$; initialize query embedding $\hat{\boldsymbol{q}}_S$
5: **for** rounds $k = 1$ **to** $K$ **do**
6:    **for** replicas $r = 1$ **to** $R$ **do**
7:       `# Replica-level coarse filtering`
8:       $b_{\boldsymbol{q}}^* \leftarrow \arg\max_b \Phi_{\mathbf{w}_r}(\hat{\boldsymbol{q}}_S)^\top \boldsymbol{o}_{b,r}$ `(closest centroid)`
9:       $\mathcal{C}_{r,0} \leftarrow \cup_{\boldsymbol{q} \in Q} \text{InvInd}^{(r)}(b_{\boldsymbol{q}}^*)$
10:      `# Replica-level centroid pruning`
11:      Compute $\sum_{\boldsymbol{q} \in Q} \max_{\boldsymbol{o} \in \text{FwdInd}^{(r)}(c)} \Phi_{\mathbf{w}_r}(\hat{\boldsymbol{q}}_S)^\top \boldsymbol{o}$, for every corpus item in filtered set: $c \in \mathcal{C}_{r,0}$
12:      Discard items with scores $< \tau$
13:      Retain top $n$ into $\mathcal{C}_{r,1}$
14:   `Replica Pooling:` $\mathcal{C}_1 \leftarrow \cup_{r=1}^R \mathcal{C}_{r,1}$
15:   `# Fine-grained filtering`
16:   Compute $\sum_{\boldsymbol{q} \in Q} \max_{r \in [R]} \max_{\boldsymbol{o} \in \text{FwdInd}^{(r)}(c)} \Phi_{\mathbf{w}_r}(\hat{\boldsymbol{q}}_S)^\top \boldsymbol{o}$, for $c \in \mathcal{C}_1$ and retain top $n/4$ into $\mathcal{C}_2$
17:   `# Residual scoring`
18:   For each $c \in \mathcal{C}_2$, compute $T_c = \{\hat{\boldsymbol{x}}_{\boldsymbol{\Delta}} = \Delta + \boldsymbol{o}\}$ where $(\boldsymbol{o}, \boldsymbol{\Delta}) \in \text{FwdInd}^{(r)}(c)$
19:   Compute $\mathcal{C}_3$ consisting of top-$n'$ items from $\sum_{\boldsymbol{q} \in Q} \max_{r \in [R]} \max_{\hat{\boldsymbol{x}}_{\boldsymbol{\Delta}} \in T_c} \Phi_{\mathbf{w}_r}(\hat{\boldsymbol{q}}_S)^\top \hat{\boldsymbol{x}}_{\boldsymbol{\Delta}}$
20:   `# Full-precision scoring+selection`
21:   $c_k \leftarrow \arg\max_{c \in \mathcal{C}_3} F(c \mid S, Q)$; $S_k \leftarrow S_{k-1} \cup \{c_k\}$
22:   Update state-dependent query representation $\hat{\boldsymbol{q}}_{S_k}$
23: **return** $S_K$

---

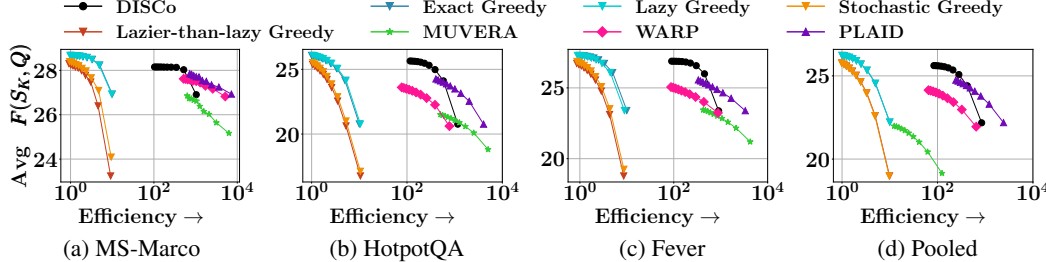

Figure 2: Trade off between efficiency and average coverage objective of DISCo and state-of-the-art baselines on four datasets: (a) MS-Marco, (b) HotpotQA, (c) Fever, and (d) Pooled. DISCo achieves the best trade-off, matching greedy baselines in coverage while being impressively faster ($> 100$X in some cases), while consistently outperforming IR baselines in terms of average coverage objective. Efficiency is in log-scale. Upper right corner is the best quadrant.

at the *last rank*, or, for more robust aggregation over queries, use a recall-sensitive measure such as MAP. MRR is inappropriate, because it cares only about the *first relevant rank*, and, for larger $|S_{gold}|$, even NDCG will not care about recalling all gold items. Additional performance measures are discussed in Appendix I.1.

In all experiments, we choose $K = 10$. Both Stochastic Greedy and Lazier-than-lazy Greedy restrict their search on a subset $\mathcal{C}'$ of corpus items selected uniformly at random, with $|\mathcal{C}'| = |\mathcal{C}| \log(1/\varepsilon')/K$. We chose $\varepsilon' = 0.5$. More details are in Table 3, Figure 4 and Appendix H.7.

## 4.1 RESULTS

**Trade-off between coverage and efficiency** Figure 2 shows the trade-off between average coverage objective $\overline{F}_K$ and the efficiency with respect to Exact Greedy, obtained by varying the subset size $K$. We make the following observations: **(1)** DISCo trades off between the mean coverage value $\overline{F}_K$ and the average query time more effectively than all the baselines. **(2)** DISCo is significantly more efficient than the variants of the greedy algorithm, *viz.*, Exact Greedy, Lazy Greedy, Stochastic Greedy and Lazier-than-lazy Greedy. In the MS-Marco dataset, DISCo is at least **100**× faster than these variants. **(3)** IR baselines, *e.g.*, PLAID, MUVERA and WARP are highly efficient, owing to their indexing and scalable search pipeline. PLAID shows the best performance among these IR methods, due to their multivector approach, while MUVERA performs the worst with their single vector approach. **(4)** Stochastic Greedy and Lazier-than-lazy Greedy show a poor trade-off due to query agnostic randomized pruning.

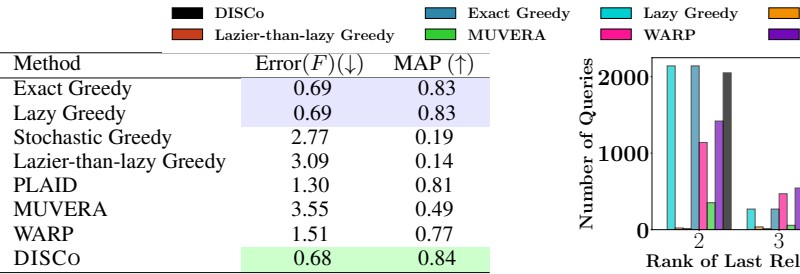

| Method | Error$(F)$($\downarrow$) | MAP ($\uparrow$) |
|---|---|---|
| Exact Greedy | 0.69 | 0.83 |
| Lazy Greedy | 0.69 | 0.83 |
| Stochastic Greedy | 2.77 | 0.19 |
| Lazier-than-lazy Greedy | 3.09 | 0.14 |
| PLAID | 1.30 | 0.81 |
| MUVERA | 3.55 | 0.49 |
| WARP | 1.51 | 0.77 |
| DISCo | 0.68 | 0.84 |

Table 3: Comparison of DISCo with baselines on gold labels of HotpotQA ($|S_{gold}| = 2$), in terms of Error$(F) = \sum_{Q \in \mathcal{Q}} |F(S_{gold}, Q) - F(S_K, Q)|/|\mathcal{Q}|$ with $K = 2$ and Mean Average Precision (MAP). Green (Blue) shows the (second) best performer.

Figure 4: Histogram of the rank of the last ($|S_{gold}|$-th) true relevant item, retrieved for queries in HotpotQA dataset. DISCo retrieves almost all items within 2 (sometimes 3) rounds, similar to deterministic greedy variants.

**Does $F(S, Q)$ reward $S_{gold}$?** We evaluate the suitability of the coverage objective $F(S, Q)$ in relation to the annotated ground-truth relevant items $S_{gold}$ in HotpotQA. In the first evaluation, we report on two metrics: **(1)** Error$(F)$: This is defined as the average deviation between the coverages on the gold set $S_{gold}$ and the retrieved set $S_K$ with $K = 2$, *i.e.*, Error$(F) = \sum_{Q \in \mathcal{Q}} |F(S_{gold}, Q) - F(S_2, Q)|/|\mathcal{Q}|$. **(2)** Mean Average Precision (MAP): We compute MAP on a ranked list of retrieved items, using rank as the selection order for greedy methods and MaxSim ranks for IR baselines. In Table 3, we summarize the results. We make the following observations: **(1)** DISCo achieves the lowest Error$(F)$ indicating that the coverage of the retrieved

sets closely approximates that of the gold set. It also achieves the highest MAP, indicating strong alignment of its computed subset with the gold annotations. While these numbers are comparable to Exact Greedy and Lazy Greedy, the efficiency of DISCO is significantly better (see: Figure 2). **(2)** Stochastic Greedy and Lazier-than-lazy Greedy perform worst among all methods, owing to their query-agnostic random pruning strategy.

In the second evaluation, we analyze the ranking of the *last relevant item* across methods by plotting a histogram of the position of the last gold retrieved item. A low value of the rank is better as it implies that the all items in $S_{\text{gold}}$ is retrieved within a small cutoff value of $K$. Figure 4 shows the results: DISCO more frequently retrieves all the gold documents within the top 2–3 positions, similar to Exact Greedy and Lazy Greedy, whereas IR methods such as MUVERA, PLAID, and WARP more frequently place the last relevant item at lower ranks.

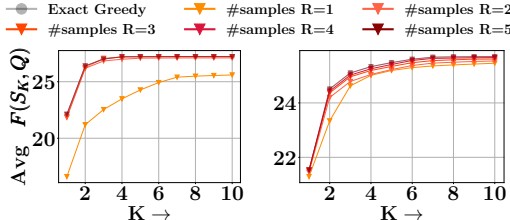
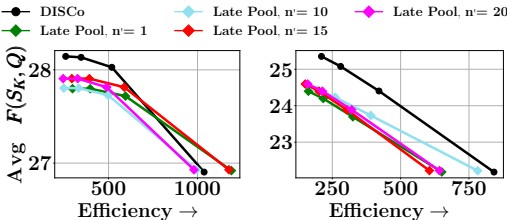

Figure 5: Coverage with approximate marginal gain. (left: NFCorpus, right: Writing)

Figure 6: Ablation study: Early vs Late pooling. (left: MS-Marco, right: Pooled)

**Quality of approximate marginal gain**   Here, we assess the quality of $G_{\mathbf{w}_{1:R}}(c, S, Q)$, our proposed approximation of the marginal gain $F(c \mid S, Q)$ in Eq. (8). To this aim, we run Algorithm 2 and compare the corresponding coverage values against that of Exact Greedy algorithm. Figure 5 summarizes the results for different numbers of hyperplanes $R$. We make the following observations. **(1)** Approximation quality improves with larger values of $R$, pushing the coverage trajectory upward. **(2)** Even with a smaller value of $R = 5$, the coverage from $G_{\mathbf{w}_{1:R}}$ matches with Exact Greedy.

**Ablation on early vs late pooling**   A natural question is whether the replica-level *early* pooling in Step 14 of Algorithm 4 is at all necessary. An easier alternative of DISCO is late pooling. Here, we process $R$ projected queries $\Phi_{\mathbf{w}_r}(\hat{\boldsymbol{q}}_S)$ independently, each returning its own top-$n'$ candidates. These are then merged and reranked to retrieve the item with the highest marginal gain, bypassing the intermediate fine-grained filtering stage. Figure 6 summarizes the results. Compared to late pooling, our current early pooling approach is superior both in terms of retrieval efficiency and the value of the final coverage objective. Early pooling approximates the pooled objective $\sum_{q \in Q} \max_{r \in [R]} \max_{o \in \text{FwdInd}(r)(c)} \Phi_{\mathbf{w}_r}(\hat{q}_S)^{\top} o$ more faithfully, ensuring that the candidate set better aligns with the marginal gains defined by Eq. (8). This early aggregation allows the method to discard low-scoring candidates earlier, reducing unnecessary final score computations while retaining documents that contribute most to the true coverage objective.

## 5   CONCLUSION

Our work introduces a novel framework for collective retrieval, which explicitly seeks to optimize for coverage of query atom vectors by subsets of corpus items. Coverage maximization can be solved in a greedy algorithm. However, here, marginal gain computation across the entire corpus is computationally expensive. To tackle this, at each iteration, we construct augmented vector representations for both query and corpus items. Then, we apply a randomized projection method and represent the marginal gain in an amenable form for ANN retrieval. This allows us to develop a practical, scalable method for sublinear-time retrieval of high-coverage item subsets. Experiments demonstrate that our method trades off between coverage and efficiency more effectively than the baselines. Future work could explore alternative similarity functions, richer representations of query-document interactions, adaptive subset selection strategies, weighted coverage, and dynamic updates.

## Ethics Statement

This work raises no specific ethical concerns. We use only publicly available datasets under their respective licenses and introduce no new human-subject data. Our contribution is an algorithmic method that improves collective-coverage retrieval over open-source corpora.

## Reproducibility Statement

We have ensured that our results are fully reproducible. We release: (1) complete source code; (2) configuration files specifying all hyperparameters (e.g., number of random hyperplanes $R$, subset size $K$, pruning thresholds $\tau$, IVF cluster count $B$, quantizer settings, maximum candidates per stage, and reranking cutoffs); (3) scripts to build indices; (4) detailed environment specs; (5) random seeds; and (6) scripts to generate all plots. We provide detailed derivations and complete proofs of all theoretical results in the appendix.

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

# A Dense Subset Index for Collective Query Coverage (Appendix)

## A BROADER IMPACT

We have presented DISCO, an indexing system for collective scoring of subsets of corpus items in response to a query. This stands in contrast to traditional retrieval, where items compete against each other to occupy top-$K$ positions. Applications that use traditional retrieval, yet need collective coverage, often inflate $K$ with the hope of including all corpus items in the optimal query cover. Replacing that approach with DISCO may reduce query processing cost and carbon footprint, as well as "lost in the middle" (Liu et al., 2023) distractions for downstream LLM-based reasoning and generation modules, resulting in better end task performance as well as saved computations.

## B LIMITATIONS

We have demonstrated that DISCO is a novel and performant method for collaborative subset retrieval. However, there are avenues for improvement. We outline these below for future exploration.

**Fairness**. Our proposed coverage objective does not account for notions of fairness, e.g. group fairness or individual fairness (Zehlike et al., 2022). For benchmarks where fairness is a factor, such as (Ekstrand et al., 2023), collaborative *and* fair retrieval is a challenging and promising avenue for future work.

**Diversity**. As of now, we have focused only on coverage maximization. We do not incorporate diversity explicitly in the model However, one can subtract a self-similarity term among the corpus tokens from the coverage objective to obtain a diversity encouraged coverage optimization.

**Evolving corpora**. DISCO is not currently designed to handle evolving corpora. In all datasets, the corpus is fixed, and indexing is done once, with no further updates. Handling changing corpus sets would be an interesting follow-up.

Addressing these limitations will improve the real-world readiness and deployment capability of DISCO.

## C LLM USAGE

We used an LLM strictly and only for (1) ancillary writing support for correcting grammar, suggesting alternative phrasing, and (2) very occasionally, literature search. No LLM was used to generate ideas, design experiments, analyze data, implement algorithms, or produce results. Any model-suggested wording, URL, or citation was reviewed and further revised by the authors.

## D FURTHER MOTIVATING SCENARIOS

### D.1 MULTI-HOP QA

We gave an example of collective passage retrieval for multihop QA in Section 1. In recent years, retrieval is often followed by a large language model (LLM) that reads and encodes the retrieved passages, then decodes the answer — in other words, such systems implement retrieval augmented generation (RAG) (Guu et al., 2020; Lewis et al., 2021; Ram et al., 2023). It has been reported that the end-to-end performance of the RAG search system is sensitive to the position of the answer-bearing passages in the LLM's input context (Liu et al., 2023). Adding irrelevant passages in the context can also be deleterious, leading to incorrect answers. Therefore, high recall of all passages needed to infer the answer, within a tight budget $K$, may enhance downstream generation. This realization is gaining ground in the NLP community. One recent response has been to resort to an LLM-based solution to subset selection, based on verbal instruction (Lee et al., 2025).

### D.2 KNOWLEDGE GRAPH QA

QA over KGs like Wikidata take two forms: semantic interpretation (Berant et al., 2013), where a natural language question is translated into a structured query, and subgraph retrieval and answer generation (Saxena et al., 2020; Sun et al., 2024). To enable LLMs to deal with billion-node graphs,

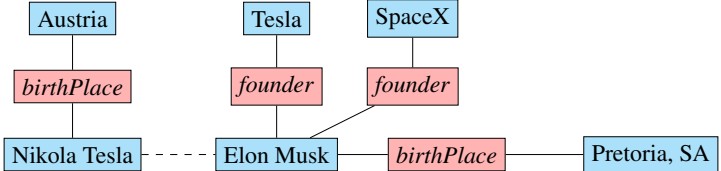

Figure 7: Example of KGQA taken from Dubey et al. (2018), for the query "where was the founder of Tesla and SpaceX born?" Observe that there are two text matches with *Tesla*, one leading to 'Austria' as the (incorrect) answer, but the correct answer node must be close to both Tesla and SpaceX. If edges are visualized as 'passages', a collective retrieval system must score highly the right subgraph, but not the left one.

subgraph retrieval is a critical step in the latter approach. As Figure 7 shows, the goal is to retrieve a subgraph that *collectively* contains the evidence of partial match with the question, and evidence in favor and answer node(s) or edge(s). Allowing nodes or edges to *compete* at matching the whole question would be misguided.

In the table below, the aggregate contractual principal amount of loans on nonaccrual status and/or more than 90 days past due (which excludes loans carried at zero fair value and considered uncollectible) exceeds the related fair value primarily because the firm regularly purchases loans, such as distressed loans, at values significantly below the contractual principal amounts:

| ($ in millions) | As of December | |
| --- | --- | --- |
| | 2016 | 2015 |
| **Performing loans and long-term receivables** | | |
| Aggregate contractual principal in excess of fair value | $ 478 | $ 1,330 |
| **Loans on nonaccrual status and/or more than 90 days past due** | | |
| Aggregate contractual principal in excess of fair value | 8,101 | 9,600 |
| Aggregate fair value on loans on nonaccrual status and/or more than 90 days past due | 2,138 | 2,391 |

The table below presents information about our funding sources.

| $ in millions | As of December | | | |
| --- | --- | --- | --- | --- |
| | **2018** | | **2017** | |
| Deposits | $158,257 | 25% | $138,604 | 23% |
| **Collateralized financings:** | | | | |
| Repurchase aggrements | 78,723 | 13% | 84,718 | 14% |
| Securities loaned | 11,808 | 2% | 14,793 | 2% |
| Other secured financings | 21,433 | 3% | 24,788 | 2% |
| Total collateralized financings | 111,964 | 18% | 124,299 | 20% |
| Unsecured short-term borrowings | 40,502 | 7% | 46,922 | 8% |
| Unsecured long-term borrowings | 224,149 | 36% | 217,687 | 36% |
| Total shareholders' equity | 90,185 | 14% | 82,243 | 13% |
| Total funding sources | $625,057 | 100% | $609,755 | 100% |

Question: What is the sum of securities loaned in 2017 and aggregate contractual principal in excess of fair value in 2015 (in millions)?

Figure 8: Example of table QA taken from Kumar et al. (2025). Note that the question has poor match or coverage by any single table element, but there is a small collection $S$ of table elements that collectively cover large (colored) spans of the question. Current practice linearizes such tables into text for LLMs, polluting the match scores with much extraneous noise from irrelevant parts of the table.

### D.3 TABLE QA

Another motivation comes from QA in the domain of textual tables (Chen et al., 2022; Zhao et al., 2022c; Kumar et al., 2025), such as appear in documents for human consumption. As Figure 8 illustrates, attempting to match the whole question against the whole table would result in weak and noisy scores. In contrast, recognizing that certain spans in the question are *covered* by certain coherent table elements (that are also spatially related) results in retrieving values from cells, from which answers can be reasoned out.

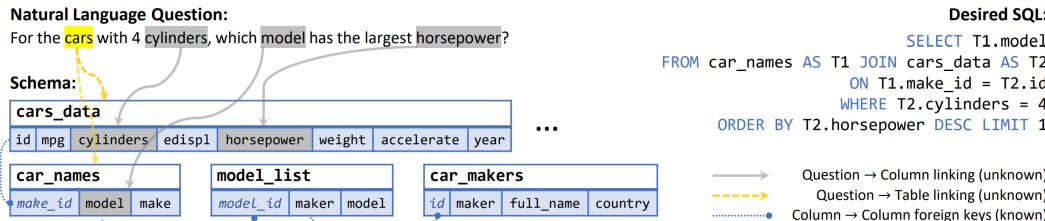

Figure 9: Example of text2sql taken from Wang et al. (2021). Schema retrieval involves mapping question spans to schema elements (table and column names), after which SQL can be generated.

### D.4 SCHEMA RETRIEVAL FOR TEXT2SQL

Text2sql is a form of semantic interpretation where the target database is relational (Katsogiannis-Meimarakis et al., 2023). As with text QA, KGQA and table QA, LLMs have gained use as semantic translators and interpreters also for text2sql (Liu et al., 2025b). Enterprise schema can contain thousands of tables and tens of thousands of columns. The schema can be represented as a graph where nodes represent tables and their columns (Wang et al., 2021), with primary-key-foreign-key and joinability relations connecting column nodes. Providing the full schema along with the natural language question as context does not only tax GPU usage in modern decoder-only LLMs, but also risks the LLM not being able to sift the schema for the relevant parts (Liu et al., 2023). As shown in Figure 9, tables and columns have to be selected to *collectively* cover specific spans in the question. Once again, individual schema elements are poor matches for the question, and it would be incorrect to pit them for relevance against each other. Unlike in passage retrieval, the total schema size may be small enough to formulate schema subgraph retrieval as a mixed integer linear program (Chen et al., 2025b), but this again clearly demonstrates the importance of collective scoring.

## E EXTENDED DISCUSSION OF RELATED WORK

Our work is related to submodular functions, data subset selection, ANN search, *etc*. In the following, we first contrast approaches to diversity in information retrieval to our method, and then briefly review and provide an extended discussion on related work.

### E.1 HOW OUR METHOD DIFFERS FROM DIVERSIFICATION IN IR

In continuation to the discussion in Section 3.2, we note that our focus is on coverage in the first stage itself, where we design indexing and retrieval method tailored specifically for coverage maximization. Note that, submodular maximization has been widely used since 1978, but our work focus on designing ANN retriever for coverage based submodular maximization. Therefore, our work focus on indexing and search, whereas these existing works, albeit related, focus on suitable submodular scoring function computation and the application of greedy variants to maximize it. Extensive search reveals a paucity on direct first-stage dense retrievers that optimize a query-coverage objective. A notable exception is in the use of pseudo-relevance feedback in dense retrieval to improve facet/subtopic coverage (Yu et al., 2021) — like key-value memory networks (Miller et al., 2016), they also perform multi-round dense query modification, but there are no formal coverage guarantees.

### E.2 SUBMODULAR FUNCTIONS

Submodular functions are functions that satisfy the property of diminishing return– a list of classic readings (Fujishige, 2005; Narayanan, 1997; Edmonds, 1970) provides a comprehensive discussion. They are discrete analogue of concave functions. They are used in wide variety of applications, *e.g.*, information cascade in social media (Kempe et al., 2003; Tang et al., 2015; Leskovec et al., 2007; Goyal et al., 2011; Borgs et al., 2014), document summarization (Lin et al., 2011), product recommendation (Tschiatschek et al., 2016; 2018), image processing (Jegelka et al., 2011), probablistic modeling (Gillenwater et al., 2014; Kulesza, 2012b; Gotovos et al., 2015; Iyer et al., 2015), *etc*.

### E.3 DATA SUBSET SELECTION

Our work is connected to traditional data subset selection problem, where the goal is to select data subset from a large set. It has several applications, *e.g.*, active learning (Wei et al., 2015), data

efficient learning (Mirzasoleiman et al., 2020; Killamsetty et al., 2021a; Durga et al., 2021) for training, *etc*. In active learning the goal is to obtain the labels of a unlabeled data, so that the model trained with those subset of labeled data, generalizes well (Wei et al., 2015). In data efficient training, we are given full labeled data and the goal is go select a subset of training data, so that we can achieve fast training with minimal accuracy loss (Killamsetty et al., 2021b;a; Durga et al., 2021). They typically frame it as joint optimization of parameter and subset selection, where the subset selection problem is often submodular or approximately submodular. Another line of works adopt different heuristics for data subset selection, e.g., entropy guided data subset selection (Na et al., 2021), use of another proxy model (Coleman et al., 2020). Zhou et al. (2020; 2021) provide adaptive subset selection based on curriculum learning. Our work is also related to coreset selection, where the goal is to select some subset (with or without weights) that is representative of the entire set (Feldman, 2020; Mirzasoleiman et al., 2020; Har-Peled et al., 2004). Our work aims to select a subset of items from a large set of items in an information retrieval setting, where the corpus is fixed for different queries. This allows us to perform one time preprocessing of corpus items, whose cost is amortized over large number of queries. The above applications, however, do not operate on this setting and therefore, they cannot use ANN algorithm for such selection.

### E.4 Approximate Nearest Neighbor (ANN) Search

In a nutshell, we seek to design an ANN method for marginal gain maximization within the framework of the greedy algorithm. Classical information retrieval systems used bag of words, which used token-id level matching, followed by an inverted index with TF-IDF or BM25 term weighting. Lucene (Gospodnetic et al., 2010) and Elastic search (Gormley et al., 2015) use similar techniques. Subsequently, neural models have been developed to represent tokens or atoms using contextual representations *e.g.*, BERT (Devlin et al., 2018), T5 (Raffel et al., 2020), *etc*. Documents can be represented as both single vector and multivectors of token representations. Such representations are key to dense vector retrieval, HNSW (Malkov et al., 2018; Simhadri et al., 2023), IVF (Douze et al., 2024), LSH (Charikar, 2002). Khattab et al. (2020); Santhanam et al. (2021; 2022) and our current method use multivector-based indexing using IVF. DiskANN and LSH used single vector-based indexing and retrieval. Our random projection method uses some technical results from LSH (Charikar, 2002; Jain et al., 2010).

In contrast to the above dense retrieval, sparse retrieval uses tokenized representations to build inverted index. Recently Formal et al. (2021) also proposed a token expansion method for enhanced sparse retrieval. As shown in the recent paper (Weller et al., 2025), multivector representations often results in enhanced performance as compared to single vector representations.

The single-round retrieval of REALM (Guu et al., 2020) and RAG (Lewis et al., 2021) soon gave way to multi-round RAG (Yang et al., 2024; Yu et al., 2024), sometimes with filtered retrieved items progressively padded with the query (Khattab et al., 2021), and later with more sophisticated planners extracting structured information from retrieved passages (Li et al., 2025; Liu et al., 2025a). Despite the superficial similarity of multi-round retrieval, there are critical differences between these and DISCO. While DISCO iteratively targets query coverage, effectively *canceling out* parts of the query (which remains a fixed-length vector) already covered, these are *query expansion* techniques, turning multi-hop queries into a series of exploratory document-to-document expansions.

### E.5 Diversity and fairness in selection and ranking

Another community that needs to score item subsets rather than individual items is concerned with the diversity and fairness of the returned responses. The need for diversity was first felt in the retrieval community, in the face of near-duplicate documents that (necessarily) had similar relevance to the query (Wu et al., 2024), but inspection of one document obviated the need to inspect the other.

Early methods like max marginal relevance (Carbonell et al., 1998) chose the next item based on a combination of its relevance to the query and dissimilarity to items already selected. Other techniques (Singh et al., 2018) involved solving relaxed integer linear programs with $N^2$ variables, where $N$ is the number of items in the corpus, so these cannot easily be applied as a first-stage ranking system (Guo et al., 2022).

User attention to the top-$K$ rank positions has been instrumented by search engine providers and found to steeply decrease with rank. A closely related concern to diversity is fairness, in the face

of such steep attention discounts, i.e., ensuring that user attention to an item is proportional to some intrinsic utility of the item for a query, or that user exposure of any designated group does not vary excessively across groups (Morik et al., 2020; Kim et al., 2025). These concerns are quite distinct from our coverage motive.

We note that diversity and fairness may *incidentally* favor query coverage, but these are very unreliable indirect mechanisms. It is possible to maximize diversity and relevance, yet fail to provide collective coverage of the query.

# F   PROOFS OF TECHNICAL RESULTS

## F.1   $F(\cdot, Q)$ IS MONOTONE SUBMODULAR

**Theorem 5** (Monotonicity and submodularity). *The set function $F(S, Q)$ is monotone and submodular.*

**Proof**   From Proposition 1, we observe:

$$F(c \mid S, Q) = \sum_{\boldsymbol{q} \in Q} \max_{\boldsymbol{x} \in X_c} \left[ \boldsymbol{q}^\top \boldsymbol{x} - \max_{u \in S} \max_{\boldsymbol{x}' \in X_u} \boldsymbol{q}^\top \boldsymbol{x}' \right]_+ \tag{10}$$

Clearly the marginal gain $F(c \mid S, Q) \geq 0$, which proves that $F(S, Q)$ is monotone in $S$.

Now suppose $S \subseteq T$, then we have:

$$\max_{u \in S} \max_{\boldsymbol{x}' \in X_u} \boldsymbol{q}^\top \boldsymbol{x}' \leq \max_{u \in T} \max_{\boldsymbol{x}' \in X_u} \boldsymbol{q}^\top \boldsymbol{x}' \tag{11}$$

$$\implies \sum_{\boldsymbol{q} \in Q} \max_{\boldsymbol{x} \in X_c} \left[ \boldsymbol{q}^\top \boldsymbol{x} - \max_{u \in S} \max_{\boldsymbol{x}' \in X_u} \boldsymbol{q}^\top \boldsymbol{x}' \right]_+ \geq \sum_{\boldsymbol{q} \in Q} \max_{\boldsymbol{x} \in X_c} \left[ \boldsymbol{q}^\top \boldsymbol{x} - \max_{u \in T} \max_{\boldsymbol{x}' \in X_u} \boldsymbol{q}^\top \boldsymbol{x}' \right]_+. \tag{12}$$

From the formula of marginal again in Eq. (10), we have: $F(S, Q) \geq F(T, Q)$.

## F.2   PROOF OF PROPOSITION 1

**Proposition 1.** *Given the coverage objective $F(S, Q)$ defined in Eq. (2), the marginal gain is:*

$$F(c \mid S, Q) = \sum_{\boldsymbol{q} \in Q} \max_{\boldsymbol{x} \in X_c} \left[ \boldsymbol{q}^\top \boldsymbol{x} - \max_{u \in S} \max_{\boldsymbol{x}' \in X_u} \boldsymbol{q}^\top \boldsymbol{x}' \right]_+ \tag{13}$$

**Proof**   We assume that $c \notin S$ to prove that:

$$
\begin{aligned}
F(c \mid S) &= F(S \cup \{c\}, Q) - F(S, Q) \\
&= \sum_{\boldsymbol{q} \in Q} \max_{\boldsymbol{x} \in \cup_{u \in S \cup \{c\}} X_u} \boldsymbol{q}^\top \boldsymbol{x} \ - \ \sum_{\boldsymbol{q} \in Q} \max_{\boldsymbol{x} \in \cup_{u \in S} X_u} \boldsymbol{q}^\top \boldsymbol{x} \\
&= \sum_{\boldsymbol{q} \in Q} \left[ \max_{u \in S \cup \{c\}} \left( \max_{\boldsymbol{x} \in X_u} \boldsymbol{q}^\top \boldsymbol{x} \right) - \max_{u \in S} \left( \max_{\boldsymbol{x} \in X_u} \boldsymbol{q}^\top \boldsymbol{x} \right) \right] \\
&= \sum_{\boldsymbol{q} \in Q} \left[ \max \left( \max_{\boldsymbol{x} \in X_c} \boldsymbol{q}^\top \boldsymbol{x}, \max_{u \in S} \left( \max_{\boldsymbol{x} \in X_u} \boldsymbol{q}^\top \boldsymbol{x} \right) \right) - \max_{u \in S} \left( \max_{\boldsymbol{x} \in X_u} \boldsymbol{q}^\top \boldsymbol{x} \right) \right]
\end{aligned}
\tag{14}
$$

Letting $a = \max_{\boldsymbol{x} \in X_c} \boldsymbol{q}^\top \boldsymbol{x}$ and $b = \max_{u \in S} \left( \max_{\boldsymbol{x} \in X_u} \boldsymbol{q}^\top \boldsymbol{x} \right)$, we apply the identity $\max(a, b) - b = \max(0, a - b)$ to the term inside the summation:

$$= \sum_{\boldsymbol{q} \in Q} \max \left( 0, \max_{\boldsymbol{x} \in X_c} \boldsymbol{q}^\top \boldsymbol{x} - \max_{u \in S} \left( \max_{\boldsymbol{x} \in X_u} \boldsymbol{q}^\top \boldsymbol{x} \right) \right)$$

Using the notation $[z]_+ = \max(0, z)$ for the positive part of $z$, we arrive at the compact final form:

$$= \sum_{\boldsymbol{q} \in Q} \left[ \max_{\boldsymbol{x} \in X_c} \boldsymbol{q}^\top \boldsymbol{x} - \max_{u \in S} \left( \max_{\boldsymbol{x} \in X_u} \boldsymbol{q}^\top \boldsymbol{x} \right) \right]_+ \tag{15}$$

## F.3   PROOFS FOR RANDOM HYPERPLANE APPROXIMATION

**Theorem 2.** *Given any two vectors $\boldsymbol{u}, \boldsymbol{v} \in \mathbb{R}^{d+1}$, let $\mathbf{w}$ be a random hyperplane $\mathbf{w} \sim \mathcal{N}(0, \boldsymbol{I}_{d+1})$, and $\Phi_{\mathbf{w}}$ be the transformation defined in Eq. (6). Then, we have the following result:*

$$\Phi_{\mathbf{w}}(\boldsymbol{u})^\top \Phi_{\mathbf{w}}(\boldsymbol{v}) = [\boldsymbol{u}^\top \boldsymbol{v}]_+ \qquad \text{with probability } p \geq 0.5. \tag{16}$$

**Proof**   We show that: $p = 1 - \frac{1}{\pi} \cos^{-1} \left( \frac{|\boldsymbol{u}^\top \boldsymbol{v}|}{||\boldsymbol{u}|| \cdot ||\boldsymbol{v}||} \right) \geq 0.5$. The equality here is tight, when $|\boldsymbol{u}^\top \boldsymbol{v}| = 0$. We shall make use of the well known result when $\mathbf{w}$ is sampled from a spherically

symmetric distribution (Charikar, 2002):

$$\mathbb{P}(\text{sign}(\mathbf{w}^\top \boldsymbol{u}) = \text{sign}(\mathbf{w}^\top \boldsymbol{v})) = 1 - \frac{1}{\pi} \cos^{-1}\left(\frac{\boldsymbol{u}^\top \boldsymbol{v}}{||\boldsymbol{u}|| \cdot ||\boldsymbol{v}||}\right) \tag{17}$$

where, $\text{sign} : \mathbb{R} \to \{-1, +1\}$ is the sign function.

Let $s = \Phi_{\mathbf{w}}(\boldsymbol{u})^\top \Phi_{\mathbf{w}}(\boldsymbol{v}) = \frac{1}{2}(\boldsymbol{u}^\top \boldsymbol{v} + \text{sign}(\mathbf{w}^\top \boldsymbol{u}) \cdot \text{sign}(\mathbf{w}^\top \boldsymbol{v}) \cdot \boldsymbol{u}^\top \boldsymbol{v})$

**Case 1:** $\boldsymbol{u}^\top \boldsymbol{v} > 0$. This also implies angle between $\boldsymbol{u}$ and $\boldsymbol{v}$ is less than $\pi/2$.

In this case, $[\boldsymbol{u}^\top \boldsymbol{v}]_+ = \boldsymbol{u}^\top \boldsymbol{v}$. Meanwhile, $s = \boldsymbol{u}^\top \boldsymbol{v}$ iff $\text{sign}(\mathbf{w}^\top \boldsymbol{u}) = \text{sign}(\mathbf{w}^\top \boldsymbol{v})$ and $s = 0$ otherwise.

$$\Pr(s = [\boldsymbol{u}^\top \boldsymbol{v}]_+) = \Pr(s = \boldsymbol{u}^\top \boldsymbol{v}) \tag{18}$$

$$= \mathbb{P}(\text{sign}(\mathbf{w}^\top \boldsymbol{u}) = \text{sign}(\mathbf{w}^\top \boldsymbol{v})) \tag{19}$$

$$= 1 - \frac{1}{\pi} \cos^{-1}\left(\frac{\boldsymbol{u}^\top \boldsymbol{v}}{||\boldsymbol{u}|| \cdot ||\boldsymbol{v}||}\right) \tag{20}$$

$$\geq \frac{1}{2} \quad \text{(Acute angle between } \boldsymbol{u} \text{ and } \boldsymbol{v}) \tag{21}$$

**Case 2:** $\boldsymbol{u}^\top \boldsymbol{v} < 0$. This also implies angle between $\boldsymbol{u}$ and $\boldsymbol{v}$ is more than $\pi/2$.

In this case, $[\boldsymbol{u}^\top \boldsymbol{v}]_+ = 0$. Meanwhile, $s = 0$ if $\text{sign}(\mathbf{w}^\top \boldsymbol{u}) = -\text{sign}(\mathbf{w}^\top \boldsymbol{v})$ and $s = \boldsymbol{u}^\top \boldsymbol{v} < 0$ otherwise.

$$\Pr(s = [\boldsymbol{u}^\top \boldsymbol{v}]_+) = \Pr(s = 0) \tag{22}$$

$$= \mathbb{P}(\text{sign}(\mathbf{w}^\top \boldsymbol{u}) = -\text{sign}(\mathbf{w}^\top \boldsymbol{v})) \tag{23}$$

$$= \frac{1}{\pi} \cos^{-1}\left(\frac{\boldsymbol{u}^\top \boldsymbol{v}}{||\boldsymbol{u}|| \cdot ||\boldsymbol{v}||}\right) \tag{24}$$

$$\geq \frac{1}{2} \quad \text{(Obtuse angle between } \boldsymbol{u} \text{ and } \boldsymbol{v}) \tag{25}$$

**Case 3:** $\boldsymbol{u}^\top \boldsymbol{v} = 0$: In that case, $s = 0 = [\boldsymbol{u}^\top \boldsymbol{v}]_+$ with probability one.

Combining these cases, the probability that $s = [\boldsymbol{u}^\top \boldsymbol{v}]_+$ is $p \geq \frac{1}{2}$. Furthermore, note that $s$ is always $\leq [\boldsymbol{u}^\top \boldsymbol{v}]_+$. ∎

**Theorem 3.** *Suppose $\{\mathbf{w}_1, \ldots, \mathbf{w}_R\}$ be $R$ random hyperplanes with $\mathbf{w}_r \overset{i.i.d.}{\sim} \mathcal{N}(0, \boldsymbol{I}_{d+1})$. Given the state-dependent representation for each query token $\widehat{\boldsymbol{q}}_S := [\boldsymbol{q}; F(S, \boldsymbol{q})]$ and the query agnostic representation of each corpus token $\widehat{\boldsymbol{x}} := [\boldsymbol{x}; -1]$ as defined in Eq. (4), let the randomized function $G_{\mathbf{w}_{1:R}}$ be defined as:*

$$G_{\mathbf{w}_{1:R}}(c; S, Q) = \sum_{\boldsymbol{q} \in Q} \max_{r \in [R]} \max_{\boldsymbol{x} \in X_c} \Phi_{\mathbf{w}_r}(\widehat{\boldsymbol{q}}_S)^\top \Phi_{\mathbf{w}_r}(\widehat{\boldsymbol{x}}) \tag{26}$$

*Then, for any $c \notin S$, $S \subset \mathcal{C}$ and $Q = \{\boldsymbol{q}\}$, a positive probability $\delta > 0$ and $R \geq \log(|Q|/\delta)$, we will have $G_{\mathbf{w}_{1:R}}(c; S, Q) = F(c \mid S, Q)$, with probability atleast $1 - \delta$.*

**Proof** This relies on Theorem 2.

Given a fixed (augmented) token $\widehat{\boldsymbol{q}}_S$, let $\widehat{\boldsymbol{x}}$ be the (augmented) token in $X$ that maximizes $\widehat{\boldsymbol{q}}_S^\top \widehat{\boldsymbol{x}}$ (where these are $(d+1)$-dimensional augmented vectors). Then, we can write:

$$\Phi_{\mathbf{w}_r}(\widehat{\boldsymbol{q}}_S)^\top \Phi_{\mathbf{w}_r}(\widehat{\boldsymbol{x}}) = [\widehat{\boldsymbol{q}}_S^\top \widehat{\boldsymbol{x}}]_+ \text{ with probability } p \geq \frac{1}{2}. \tag{27}$$

Eq. (27) implies that:

$$\max_{\boldsymbol{x} \in X_c} \Phi_{\mathbf{w}_r}(\widehat{\boldsymbol{q}}_S)^\top \Phi_{\mathbf{w}_r}(\widehat{\boldsymbol{x}}) = \max_{\boldsymbol{x} \in X_c} \max[\widehat{\boldsymbol{q}}_S^\top \widehat{\boldsymbol{x}}]_+ \text{ with probability } p \geq \frac{1}{2}. \tag{28}$$

Note that $[\widehat{\boldsymbol{q}}_S^\top \widehat{\boldsymbol{x}}]_+$ is the maximum value of the approximation. Hence,

$$\Phi_{\mathbf{w}_r}(\widehat{\boldsymbol{q}}_S)^\top \Phi_{\mathbf{w}_r}(\widehat{\boldsymbol{x}}) \leq [\widehat{\boldsymbol{q}}_S^\top \widehat{\boldsymbol{x}}]_+ \quad \text{for all } r \in [R] \tag{29}$$

We note that Eq. (29) implies that:

$$\max_{\boldsymbol{x} \in X_c} \Phi_{\mathbf{w}_r}(\widehat{\boldsymbol{q}}_S)^\top \Phi_{\mathbf{w}_r}(\widehat{\boldsymbol{x}}) \leq \max_{\boldsymbol{x} \in X_c} \max[\widehat{\boldsymbol{q}}_S^\top \widehat{\boldsymbol{x}}]_+ \text{ with probability } = 1 \tag{30}$$

Thus for $R$ sampled hyperplanes, the event that *maximum of the above approximations gives us the correct value* is the same as the event that *atleast one of $R$ augmentations giving the correct value*. Hence, we have:

$$\max_{r\in[R]}\max_{\boldsymbol{x}\in X_c}(\widehat{\boldsymbol{q}}_S)^\top\Phi_{\mathbf{w}_r}(\widehat{\boldsymbol{x}}) = [\widehat{\boldsymbol{q}}_S^\top\widehat{\boldsymbol{x}}]_+ \tag{31}$$

$$\iff \exists r\in[R], \quad \max_{\boldsymbol{x}\in X_c}\max_{\boldsymbol{x}\in X_c}(\widehat{\boldsymbol{q}}_S)^\top\Phi_{\mathbf{w}_r}(\widehat{\boldsymbol{x}}) = [\widehat{\boldsymbol{q}}_S^\top\widehat{\boldsymbol{x}}]_+ \tag{32}$$

Hence we have

$$\Pr\left(\max_{r\in[R]}\max_{\boldsymbol{x}\in X_c}(\widehat{\boldsymbol{q}}_S)^\top\Phi_{\mathbf{w}_r}(\widehat{\boldsymbol{x}}) = [\widehat{\boldsymbol{q}}_S^\top\widehat{\boldsymbol{x}}]_+\right)$$

$$= 1 - \prod_{r\in[R]}\left(1 - \Pr\left(\max_{\boldsymbol{x}\in X_c}(\widehat{\boldsymbol{q}}_S)^\top\Phi_{\mathbf{w}_r}(\widehat{\boldsymbol{x}}) = [\widehat{\boldsymbol{q}}_S^\top\widehat{\boldsymbol{x}}]_+\right)\right)$$

$$\geq 1 - \left(1 - \frac{1}{2}\right)^R = 1 - \frac{1}{2^R}. \tag{33}$$

Finally, applying the union bound over all query tokens in $Q$, the probability that all tokens are correctly approximated is $\geq 1 - |Q|/2^R$. Setting $1 - |Q|/2^R \geq 1 - \delta$ gives us the condition $R \geq \log(|Q|/\delta)$. ∎

This leads to the following corollary.

**Corollary 6.** *For a given set of items $S$ and query $Q$, let $G_{\mathbf{w}_{1:R}}(\cdot\,|\,S,Q)$ be as defined as in Theorem 3 with $R \geq \log(|Q|/\delta)$. Let $c_G = \arg\max_{c'\in\mathcal{C}\setminus S}(G_{\mathbf{w}_{1:R}}(c'\,|\,S,Q)$, then for the coverage function $F$,*

$$F(c_G\,|\,S,Q) = \max_{c'\in\mathcal{C}\setminus S}F(c'\,|\,S,Q) \quad w.p.\ 1-\delta \tag{34}$$

**Proof** Let $c^*$ be a document that maximizes the marginal, *i.e.* $F(c^*\,|\,S,Q) = \max_{c'\in\mathcal{C}\setminus S}F(c'\,|\,S,Q)$. Then, from Theorem 3, $G_{\mathbf{w}_{1:R}}(c^*\,|\,S,Q) = F(c^*\,|\,S,Q)$ with probability $1-\delta$.

We use the fact that $G_{\mathbf{w}_{1:R}}$ is bounded above by $F$, which means that for any document $c \in \mathcal{C}\setminus S$, $G_{\mathbf{w}_{1:R}}(c\,|\,S,Q) \leq F(c\,|\,S,Q) \leq F(c^*\,|\,S,Q)$.

As $G_{\mathbf{w}_{1:R}}(c_G\,|\,S,Q) = \max_{c'\in\mathcal{C}\setminus S}G_{\mathbf{w}_{1:R}}(c'\,|\,S,Q) \geq G_{\mathbf{w}_{1:R}}(c^*\,|\,S,Q)$, the above two conditions imply that the bound becomes tight with probability atleast $1-\delta$, *i.e.* $F(c_G\,|\,S,Q) = F(c^*\,|\,S,Q) = \max_{c'\in\mathcal{C}\setminus S}F(c'\,|\,S,Q)$. ∎

### F.4 PROOF OF APPROXIMATE GREEDY GUARANTEE

**Theorem 4.** *Let $S_K$ be the set of $K$ corpus items selected by Algorithm 2, and let $S^*$ be the optimal set for $F$, i.e., $S^* = \arg\max_S F(S,Q)$. Given $\delta \in (0,1]$, if we set the number of random projection vectors $R \geq \log(|Q|/\delta)$,*

$$\mathbb{E}_{\mathbf{w}_{1:R}}[F(S_K,Q)] \geq (1 - 1/e - \delta)\cdot F(S^*,Q). \tag{35}$$

**Proof** For notational simplicity we shall drop the subscript of the expectation. Let the set of documents selected at each iteration of the algorithm be $S_k$ for $k = 1,\ldots,K$. Let $S_{k+1} = S_k \cup \{c_{k+1}\}$. Greedy choice is $\bar{c}_{k+1}$. From Lemma 7, we have:

$$\sum_{a\in S^*\setminus S_k}F(a\,|\,S_k,Q) \geq F(S^*,Q) - F(S_k,Q) \tag{36}$$

Combining this with Lemma 7:

$$\mathbb{E}\left[F(c_{k+1}\,|\,S_k,Q)\,|\,S_k\right] \geq \frac{(1-\delta)}{K}(F(S^*,Q) - F(S_k,Q)) \tag{37}$$

Since $F(S_{k+1},Q) = F(S_k,Q) + F(c_{k+1}\,|\,S_k,Q)$, we can rewrite this as:

$$\mathbb{E}\left[F(S^*,Q) - F(S_{k+1}|Q)\,|\,S_k\right] \leq \left(1 - \frac{1-\delta}{K}\right)(F(S^*,Q) - F(S_k,Q)) \tag{38}$$

We make the following observation, as $S_0 = \{\}$ and $S_1, .., S_k, ..S_K$ are defined sequentially. Thus, for a random variable $X$, $\mathbb{E}[X] = \mathbb{E}[X \mid S_0] = \mathbb{E}[\mathbb{E}[\cdots \mathbb{E}[X \mid S_K] \cdots \mid S_1] \mid S_0]$, Now we can write

$$\mathbb{E}\left[F(S^*, Q) - F(S_K) \mid S_{K-1}\right] \leq \left(1 - \frac{1-\delta}{K}\right)(F(S^*, Q) - F(S_{K-1}, Q)) \tag{39}$$

One more round of taking expectation gives us:

$$\mathbb{E}\left[\mathbb{E}\left[F(S^*, Q) - F(S_K) \mid S_{K-1}\right] \mid S_{K-2}\right] \tag{40}$$

$$\leq \mathbb{E}\left[\left(1 - \frac{1-\delta}{K}\right)(F(S^*, Q) - F(S_{K-1}, Q)) \mid S_{K-2}\right] \tag{41}$$

$$\leq \left(1 - \frac{1-\delta}{K}\right)^2 (F(S^*, Q) - F(S_{K-2}, Q)) \tag{42}$$

By unrolling up to $S_0$,

$$\mathbb{E}\left[F(S^*, Q) - F(S_K, Q)\right] \leq \left(1 - \frac{1-\delta}{K}\right)^K (F(S^*, Q) - \underbrace{F(S_0, Q)}_{=0}) \tag{43}$$

$$= \left(1 - \frac{1-\delta}{K}\right)^K F(S^*, Q) \tag{44}$$

Rearranging terms,

$$\mathbb{E}\left[F(S_K, Q)\right] \geq \left(1 - \left(1 - \frac{1-\delta}{K}\right)^K\right) F(S^*, Q) \tag{45}$$

$$\geq \left(1 - e^{-(1-\delta)}\right) F(S^*, Q) \tag{46}$$

$$\geq (1 - 1/e - \delta) F(S^*, Q) \tag{47}$$

The last inequality uses the fact that $e^{-(1-\delta)} \leq 1/e + \delta$. This completes the proof. ∎

**Lemma 7.** *Let* $c_{k+1} = \arg\max_{c \in \mathcal{C} \setminus S_k} G_{\mathbf{w}_{1:R}}(c; S_k, Q)$, *i.e. the item returned by the approximate greedy algorithm 2 in the* $k^{th}$ *round for inclusion into* $S_k$ *to form* $S_{k+1}$. *Then, the expected marginal gain at each step satisfies:*

$$\mathbb{E}\left[F(c_{k+1} \mid S_k, Q) \mid S_k\right] \geq \frac{(1-\delta)}{K} \sum_{a \in S^* \setminus S_k} F(a \mid S_k, Q) \tag{48}$$

**Proof** Let $\bar{c}_{k+1}$ be the document maximizing the marginal gain $F(\cdot \mid S, Q)$ in $\mathcal{C} \setminus S_k$. Our Approx-greedy algorithm 2 chooses $c_{k+1} = \arg\max_{c' \in \mathcal{C} \setminus S_k} G_{\mathbf{w}_{1:R}}(c' \mid S_k, Q)$. Therefore, using Corollary 6, we have

$$F(c_{k+1} \mid S_k, Q) = F(\bar{c}_{k+1} \mid S_k, Q) \quad \text{w.p.} \geq 1 - \delta \tag{49}$$

$$\implies \mathbb{E}\left[F(c_{k+1} \mid S_k, Q) \mid S_k\right] \geq (1-\delta)F(\bar{c}_{k+1} \mid S_k, Q) \tag{50}$$

By the definition of $\bar{c}_{k+1}$, for any $a \in S^* \setminus S_k$:

$$F(\bar{c}_{k+1} \mid S_k, Q) \geq F(a \mid S_k, Q) \tag{51}$$

Therefore:

$$F(\bar{c}_{k+1} \mid S_k, Q) \geq \frac{1}{|S^* \setminus S_k|} \sum_{a \in S^* \setminus S_k} F(a \mid S_k, Q) \tag{52}$$

$$\geq \frac{1}{K} \sum_{a \in S^* \setminus S_k} F(a \mid S_k, Q) \tag{53}$$

Combining these inequalities:

$$\mathbb{E}\left[F(c_{k+1} \mid S_k, Q) \mid S_k\right] \geq \frac{(1-\delta)}{K} \sum_{a \in S^* \setminus S_k} F(a \mid S_k, Q), \tag{54}$$

which completes the proof.

## G  ADDITIONAL DETAILS ABOUT THE EXPERIMENTS

In this section, we provide the necessary details about the experiments. Our code is already uploaded in supplementary material.

### G.1  DATASETS

We evaluate on two widely used IR suites:

- **BEIR** (Thakur et al., 2021)
- **Lo**ng-**T**ail **T**opic-stratified **E**valuation (LoTTE) (Santhanam et al., 2021)

**BEIR.**  BEIR comprises heterogeneous retrieval tasks spanning multiple domains. We use the following large-corpus subsets:

| Dataset | (Test) Queries | #Documents | $\frac{\|\{Q: \|S_{\text{gold}}(Q)\|>1\}\|}{\|\mathcal{Q}\|}$ | Brief Description |
|---------|----------------|------------|------|-------------------|
| MS-Marco | 43 | 8.84M | 1.00 | Passage retrieval from web search queries (Bing) |
| HotpotQA | 7,405 | 5.23M | 1.00 | Multi-hop QA requiring evidence across documents |
| Fever | 6,666 | 5.42M | 0.12 | Claim verification with Wikipedia evidence |

Table 10: BEIR subsets used and their statistics. The fraction column is the proportion of queries with $|S_{\text{gold}}(Q)| > 1$.

**LoTTE.**  LoTTE targets out-of-domain generalization with six topic-stratified corpora constructed from Stack Exchange communities. Each corpus provides two query sets (*search* and *forum*); we use the *forum* queries derived from question titles.

| Dataset | Test Queries | #Documents | $\frac{\|\{Q: \|S_{\text{gold}}(Q)\|>1\}\|}{\|\mathcal{Q}\|}$ | Subtopics (examples) |
|---------|--------------|------------|------|----------------------|
| Lifestyle | 2,000 | 119K | 0.90 | Cooking, Sports, Travel |
| Recreation | 2,000 | 167K | 0.78 | Gaming, Anime, Movies |
| Science | 2,017 | 1.7M | 0.92 | Math, Physics, Biology |
| Technology | 2,004 | 639K | 0.95 | Apple, Android, UNIX, Security |
| Writing | 2,000 | 200K | 0.95 | English (writing, usage) |
| Pooled | 10,025 | 2.8M | 0.90 | Union of all above topics |

Table 11: LoTTE dataset. The fraction column is the proportion of queries with $|S_{\text{gold}}(Q)| > 1$.

### G.2  INDEXING STATISTICS

In Table 12, we provide statistics on index construction and memory consumption for each of the indexing based methods. We note that the memory consumption reported for DISCO is higher than for other methods due to the construction of $R = 8$ different replica indices, each housing corpus vectors that have been augmented to approximate the maximum marginal gain.

| Dataset | PLAID | MUVERA | WARP | DISCO |
|---------|-------|--------|------|-------|
| MS-Marco | 23 GB | 88 GB | 81 GB | 285 GB |
| HotpotQA | 12 GB | 52 GB | 48 GB | 172 GB |
| Fever | 17 GB | 54 GB | 87 GB | 240 GB |
| Pooled | 11 GB | 29 GB | 70 GB | 160 GB |
| Science | 5.9 GB | 18 GB | 47 GB | 88 GB |
| Technology | 2.6 GB | 6.4 GB | 14 GB | 38.9 GB |
| Writing | 0.7 GB | 2.1 GB | 3.5 GB | 11 GB |

Table 12: Index memory consumption across methods and datasets, in gigabytes (GB).

### G.3  IMPLEMENTATION DETAILS

**Embedding model**  We use transformer models trained on standard IR tasks for embedding the queries and corpus document. In particular, we use the embedding model from PLAID (Khattab et al., 2020), which is a BERT-base model finetuned on the MS-Marco dataset. The model consists of a 12-layer transformer encoder with an output dimension of 768, followed by a linear layer that projects the output embeddings down to 128 dimensions. The architecture employs WordPiece

tokenization on the raw text to generate token IDs. Following the ColBERT recipe, we L2-normalize the embeddings, and also mask out stop words in the corpus documents.

**Retrieval Engine**    We adapt our implementation starting with PLAID engine (Santhanam et al., 2022) to maximize the approximate marginal gain $G_{\mathbf{w}_{1:R}}$ which ultimately maximizes the coverage objective (2). Note that we query in parallel the $R$ replica indices in order to speed up the pipeline. We expand on this below.

Next, we describe the pertinent details of our indexing and retrieval implementation.

— *Embedding Storage:*    The mode of embedding access depends on corpus size. For smaller corpora we support in-memory storage, which is faster, while larger corpora use a disk-based mode. Embeddings are serialized during indexing in `batch_no.chunk_no.pkl` format, ensuring no embedding is duplicated across chunks. Embeddings are dumped to disk as part of the indexing process, and are then loaded either once (1) wholly in memory for memory mode, or (2) on during search on an immediate need basis in disk mode. By immediate need basis, it is meant that we iterate over the pickle files and retrieve only the requisite corpus token embeddings.

— *Augmentation:*    We sample $R$ random hyperplanes and store each in its own file to prevent random corruption. When indexing, we generate $R$ different folders, containing index data corresponding to the given hyperplane. The augmentation process takes a batch of corpus token embeddings, a hyperplane, and returns embeddings which are to be indexed only by the corresponding indexing process (out of $R$ such processes). The search procedure follows suit, in that query token embeddings are augmented on-the-fly and then passed on to the corresponding index for retrieval. Note that the data returned from the augmented indices is pooled as part of an earlier stage. The final results are returned using the non-augmented index corresponding to the given dataset.

— *Parallelization:*    We query the $R$ replica indices in parallel, in order to speed up our pipeline. In the case of DISCO, we choose multi-threading as our preferred form of parallel processing. It is known that the global interpreter lock (GIL), a data structure used to prevent parallel processing, is active in base Python installs. This limits the parallel capability of our program. However, note that in this setup no inter-process communication is required, and due to disk I/O being the chief bottleneck, we found Python's multi-threading to be faster than Python's multiprocessing module in our experiments.

In the case of late pooling, we resort to using the multiprocessing module, as no internal edits of the engine code are required.

— *Quantization:*    For quantization of the embeddings, we require that the number of embedding dimensions be such that number of quantization bits per dimension × number of dimensions is a multiple of 8 (so that the quantized vectors can be stored in bytes). As our embedding models produce 128-dimensional vectors by default, the augmentation procedure described in Section 3 would result in 129-dimensional vectors. To ensure compatibility with quantization, we take the first 127 dimensions of the embedding, which has a negligible impact on performance. For a fair comparison, we use the 127 dimensional embeddings for all methods (by truncation or setting the last dimensions to zero). We set 2 bits per dimension for quantization. We generate $R = 8$ replica indices. The quantization method is the same as PLAID.

— *Coverage Scoring:*    We perform exact scoring using efficient tensor ops. We make use of the following property: $\max_{S \cup \{c\}} f = \max(\max_S f, f(c))$ for the coverage value per token $q$. Thus, we can efficiently update and compute the vector $[F(S, q)]_{q \in Q}$. This unreduced/partially computed MaxSim score can be used for coverage scoring for our baselines, as well as for computing the augmented representations $[\widehat{q}_S]_{q \in Q}$, which is simply the concatenation of the query embeddings with the above vector.

**Hyperparameters**    While indexing, DISCO sets the number of centroids ($B$ in the indexing routine Algorithm 3) for k-means dynamically based on the number of items. k-means is computed on a much smaller subset sampled from the entire corpus before indexing. This sample is also used to estimate the total number of tokens over the entire corpus (, which is the size of the index.

The number of centroids for k-means is chosen as $\sqrt{16 \times \text{est. size}}$. However, rather than this exact quantity, we instead the largest power of 2 that is less than or equal to it, considering the bitwidth of the centroid id field. This is also the strategy used by PLAID.

During retrieval (Algorithm 4), we probe one cell per token, and we fix $n' = 1$. Subsequently, the threshold $\tau$ is set to $0.5$, and the number of documents considered after filtering is 256. These values are based on the values used by PLAID when retrieving $n'$ items.

In our ablations in Section G, we also experiment with the following combinations of hyperparameters for DISCO: $(n', n, \tau) = (10, 256, 0.5), (15, 1024, 0.45)$.

We also introduce a different variant, termed **Late Pool**, where each replica is used for end-to-end retrieval , which includes pruning, filtering and obtaining a top-$n'$. This is followed a pooling across all replicas, and the best candidate is chosen out of these. In this case, we test the same combinations of $(n', n, \tau)$, namely $(1, 256, 0.5), (10, 256, 0.5), (15, 1024, 0.45)$.

## G.4 BASELINES

We use two classes of baselines (1) Submodular Optimization based solvers $(2-4)$, and (2) Retrieval engines based on MaxSim retrieval.

**Exact Greedy** (Nemhauser et al., 1978). It performs exhaustive search over the corpus items, by computing the marginal gain $F(c \mid S_{K-1}, Q)$ for each $c \in \mathcal{C}$ for each $K$ (line 3, Algorithm 1). In this manner, it builds the solution set over $K$ iterations. The brute-force evaluation of each candidate renders it inefficient for large corpus sets. It serves as a skyline in terms of coverage performance, and is implemented without the use of any solver for this reason.

**Lazy Greedy** (Minoux, 2005). This is an accelerated variation of Exact Greedy. It builds a heap before the first iteration, which allows it to avoid exhaustive search in subsequent iterations. However, it still has to build the heap for each query, resulting in a linear time complexity. It is based on the principle that for submodular functions, the marginal gain of an item can only diminish as the algorithm progresses.

**Stochastic Greedy** (Mirzasoleiman et al., 2015). It is a variant of greedy algorithm which— instead of probing the entire corpus $\mathcal{C}$— uniformly samples a subset $\mathcal{C}' \subset \mathcal{C}$ at random to evaluate and select the next candidate. The performance of this algorithm entirely depends on the size of the subset sampled, which is in turn controlled by the $\epsilon'$ parameter. $\epsilon'$ presents a speed-coverage tradeoff.

**Lazier-than-lazy Greedy** (Mirzasoleiman et al., 2015). This variant of Exact Greedy combines the benefits of Lazy Greedy and Stochastic Greedy together. It heapifies the randomly selected subset before the first iteration, which speeds up the subsequent selections of $K - 1$ elements. The speed benefit derived may be heavily implementation dependent. In our experience, the submodlib library implements Lazier-than-lazy Greedy with the help of an `std::set` data structure, whose underlying structure is a red-black tree. This implementation choice results in overhead during execution.

**PLAID** (Santhanam et al., 2022). Here, the indexing and retrieval methods are designed for independent top-$K$ retrieval based on MaxSim score in Eq. (1). It offers multi-stage progressive pruning based retrieval, using multi-vector based IVF (Douze et al., 2024), similar to DISCO. PLAID's retrieval engine is designed to offer significant speedups over the earlier ColBERT v1/v2 (Santhanam et al., 2021; Khattab et al., 2020) style retrievers. It achieves this by clustering corpus passages into a bag of centroids and their corresponding compressed residuals.

Before doing any exact query-corpus token interactions, PLAID performs MaxSim scoring on these centroids to obtain a large pool of corpus items, and then prunes them according to a carefully chosen pruning threshold. It executes these operations using highly optimized CPU/CUDA runtimes, leading to $42.4\times$ CPU latency drop and $6.6\times$ GPU latency drop.

**MUVERA** (Dhulipala et al., 2024). It constructs fixed-dimensional single vector encodings for $Q$ and $X$, whose inner product $\epsilon$-approximates the MaxSim score (1), enabling single-vector based ANN for independent retrieval of top-$K$ items. MUVERA generates these encodings by first hashing tokens into buckets, then performing a random projection to bring bucket vectors onto a specified dimension, and finally repeating this random sketch $R$ times and concatenating the vectors. As a result, the dimension of the encoding is $d_{\text{FDE}} = B \times d_{\text{proj}} \times R$, where $B$ is the number of buckets, $d_{\text{proj}}$ is the random projection dimension, and $R$ is the number of repeats. $B$ is obtained by setting the `num_simhash_projections` parameter, which gives us $B = 2^{\text{num\_simhash\_projections}}$.

We observe that the quality of their approximation depends on $d_{\text{FDE}}$. Their experiments often use a value of $d_{\text{FDE}} = 10,240$. In our experiments, we set to $d_{\text{FDE}} = 2^5 \times 20 \times 20 = 12,800$. The MUVERA implementation provides an (optional) end-stage projection to control the final dimension size, which we leverage to obtain encodings of dimension 2560. The choices of dimensions and hyperparameters are guided by the ablation experiments in their paper.

**WARP** (Scheerer et al., 2025). It optimizes a variant of the MaxSim score, enabling the filtering of corpus tokens $x$ that are highly dissimilar to a query token $q$. This is followed by a two-stage reduction process that yields substantial latency improvements over PLAID. This baseline makes use of a finetuned T5 embedding model for its ranking, which we do not change. However, the coverage is computed using the BERT embeddings.

WARP improves over XTR (Lee et al., 2023) by replacing XTR's token residual reconstruction with the following optimizations: **(1)** Dynamic similarity imputation, with which it estimates the similarity scores of a large number of token-token pairs, **(2)** scoring using compressed residuals and avoiding residual decompression altogether, and **(3)** reducing the set of candidate passages and per-token final pairs before the last round of pooling. Additionally, the authors provide highly optimized C++ runtimes which contribute to a $41\times$ drop in latency.

Next, we describe our choice of submodular optimization solver in detail.

**—** *Submodular optimization solver:* For the submodular solver baselines, we use the `facilityLocation` solver class of the submodlib library (Kaushal et al., 2022). submodlib-based greedy programs adhere to a two stage framework. In the first stage, for the given query, a full sweep across the corpus is made to obtain pairwise scores for the similarity kernel. In the second stage, a call is made to the submodlib API with the choice of optimizer (e.g. Lazy Greedy, Lazier-than-lazy Greedy, Stochastic Greedy), $\epsilon'$ and the kernel matrix. Queries are processed in batches of 100 for the similarity computation. We note that the similarity computation contributes to a significant speed bottleneck, as seen in Figure 2, and a parallelizable algorithm for submodular optimization would be of great interest.

During initial testing, we discovered that the solution sets returned by Lazy Greedy were not matching with our Exact Greedy implementation. Additionally, Lazier-than-lazy Greedy and Stochastic Greedy were unable to perform even upto the level seen in Figure 2. Upon investigation, it was found that their implementation assumes that the kernel matrix is meant to be accessed in a column-major manner in memory. We fixed this bug by changing to the correct row-major access style. Subsequent tallying of results passed our correctness tests.

### G.5 SYSTEM CONFIGURATION

All experiments were performed on a server with seven 48GB RTX A6000 GPUs. The server has a 96-core 1.5GHz AMD Epyc CPU running Debian13. Exact Greedy tensor operations were performed on GPU, as were indexing and retrieval operations on PLAID based architectures, such as WARP and DISCo. Greedy variations such as Stochastic Greedy, Lazy Greedy and Lazier-than-lazy Greedy were implemented using the submodlib (Kaushal et al., 2022) library, and operations were performed on CPU.

## H    ADDITIONAL EXPERIMENTS

In this section, we provide results for an extensive variety of additional experiments. Specifically, we provide additional results for the relevance of the coverage objective with respect to gold items, quality of our proposed approximation to marginal gain, the tradeoff between coverage and efficiency with respect to Exact Greedy, the need for early pooling in DISCo, and various configurations of the greedy algorithms. Additionally, we provide results on the tradeoff between coverage and the number of iterations $K$, variation of hyperparameters in DISCo, and set-wise comparison with the Exact Greedy solution.

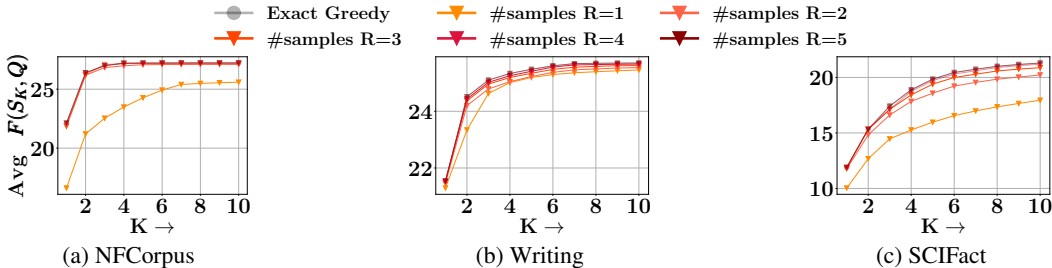

Figure 13: Coverage obtained with our approximate marginal gain formulation under varying numbers of random hyperplanes $R$. Our approximation of the marginal gain rapidly approaches the exact marginal gain skyline, with only a small number of hyperplanes needed for near-optimal coverage.

### H.1    QUALITY OF PROPOSED APPROXIMATION

We provide additional results on the assessment of the quality of the marginal gain function $G_{\mathbf{w}_{1:R}}(c, S, Q)$. Specifically, we compare the Exact Greedy algorithm (exact marginal gain at every iteration) to its random hyperplane–augmented counterpart under varying numbers of hyperplanes $R$. Figure 13 reports the coverage achieved by each configuration. We observe that as $R$ increases, the quality of the marginal gain approximation improves and the coverage rapidly approaches that of Exact Greedy, which constitutes the skyline across all datasets. Notably, only a modest number of hyperplanes is sufficient to close this gap, thereby justifying the design choice of using limited $R$ in DISCo.

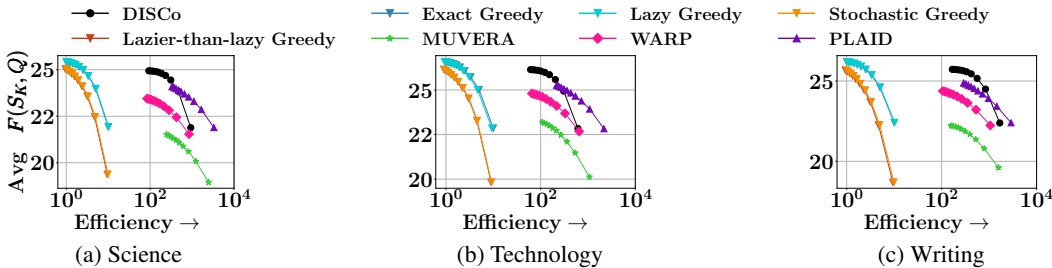

Figure 14: Analysis of tradeoff between coverage and efficiency on DISCo against state of the art baselines, across additional datasets from the LoTTE benchmark. DISCo continues to report a superior coverage-efficiency tradeoff compared to the baselines. X-axis is log-scale and upper right corner is the best performing quadrant.

### H.2    COVERAGE-EFFICIENCY TRADEOFF

We compare DISCo against the baselines on additional datasets from the LoTTE benchmark in Figure 14. We make the following observations: **(1)** DISCo continues to trade off between mean coverage and average query time more effectively. **(2)** For example, in the Technology dataset, DISCo is at least **61×** faster than the greedy variants, and at least **66×** more efficient than them. **(3)** Exact Greedy and Lazy Greedy continue to dominate coverage, but are prohibitively expensive compared to the indexing based methods. **(4)** Query agnostic randomized pruning continues to affect the performance of Stochastic Greedy and Lazier-than-lazy Greedy.

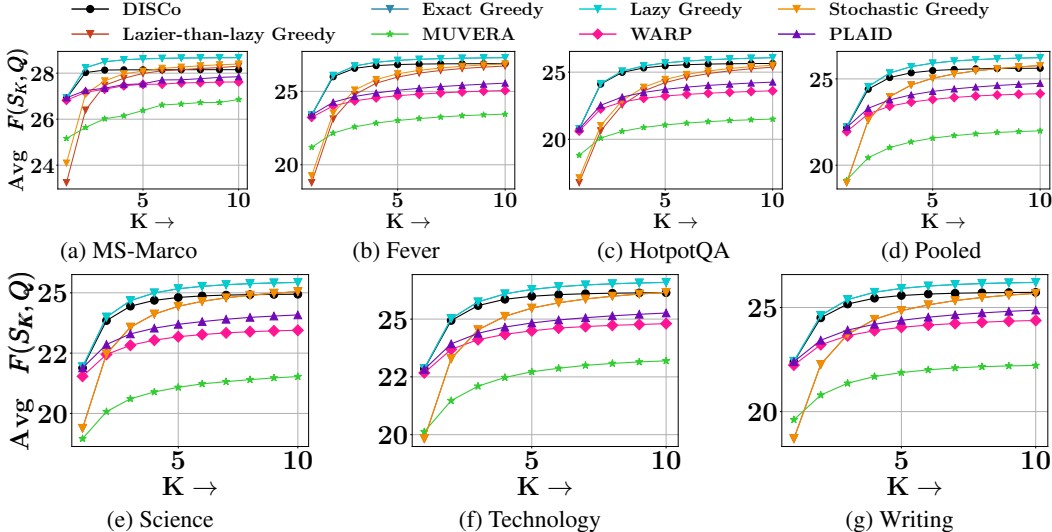

Figure 15: Analysis of coverage in terms of the upper bound $K$ on the size of solution set $S$, i.e., $|S| \leq K$. We vary $K$ from 1 to 10. Exact Greedy and Lazy Greedy provide skyline values, while DISCO consistently outperforms all other baselines.

## H.3 COVERAGE VS $K$

Here we analyze the performance of DISCO against the baselines in terms of coverage and the value of the upper bound $K$ on the size of the solution set $S$. We provide results on the MS-Marco, Fever, HotpotQA and Pooled datasets. Note that these results are independent of the time taken by any method. Figure 15 summarizes the results. Our observations are as follows. (**1**) In each of the datasets, Exact Greedy and Lazy Greedy dominate over the rest of the methods. This is explained by the fact that both methods maximize the exact marginal gain over the entire corpus set, minus the already chosen items. This is in contrast to Stochastic Greedy and Lazier-than-lazy Greedy, which sample a reduced subset of the corpus to evaluate and select the next best element at each iteration. (**2**) DISCO is consistently the next best performer. It outperforms Stochastic Greedy and Lazier-than-lazy Greedy amongst the greedy variations, and also outdoes the other indexing based methods such as PLAID, MUVERA and WARP. (**3**) PLAID is able to outperform WARP on at least three datasets, despite WARP being an extension of PLAID meant for better performance. We conclude that WARP is optimized for independent top-$K$ retrieval. (**4**) MUVERA, despite having been developed as an efficient single-vector MIPS approximator to multi-vector search, is unable to perform in the collaborative retrieval setting.

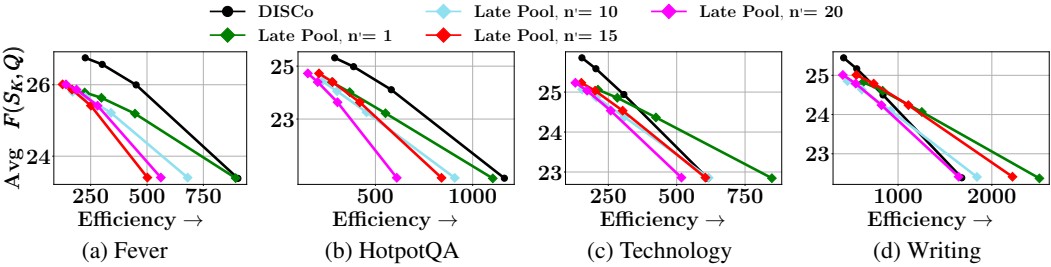

Figure 16: Ablation study to justify the architectural choice of early pooling. DISCO achieves a superior coverage-efficiency tradeoff against the late pooling baselines.

## H.4 ABLATION STUDY ON NEED FOR EARLY POOLING

We provide further results demonstrating the need for early pooling within the DISCO end-to-end retrieval pipeline. In particular, we compare DISCO with its late pooling ablations on the Fever, HotpotQA, Technology, and Writing datasets. Recall that in the late pooling variants, the $R$ projected queries $\Phi_{w_r}(\hat{q}_S)$ are processed independently via ColBERT, with each replica producing its own top-$n'$ candidates–which are merged and reranked according exact marginal gain to the cov-

erage objective $F(c \mid S, Q)$. Early pooling is the method of choice across all datasets, achieving a superior coverage-efficiency tradeoff compared to the ablations. Figure 16 summarizes the results. We observe that among the late pooling baselines, the variant with $n' = 20$ attains the best coverage, while other curves converge toward this ceiling, but all remain below the performance of early pooling.

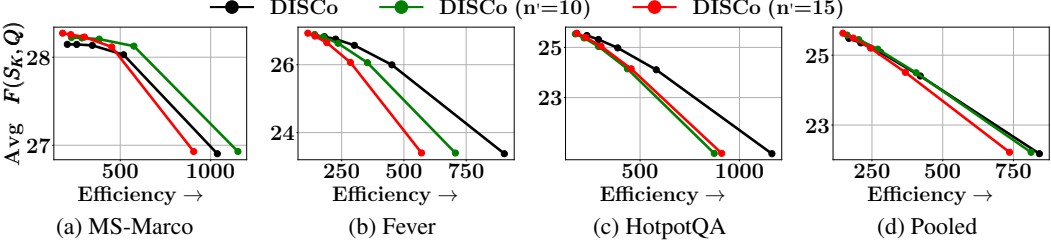

Figure 17: Ablation study on the effect of $n'$ on the performance of DISCo. While higher $n'$ provides absolute gains in coverage, this comes with a drop in efficiency. This result justifies the use of the default setting in our experiments involving DISCo.

## H.5 ABLATION STUDY ON VARYING $n'$ FOR DISCo

In this experiment, we vary the $n'$ hyperparameter for DISCo–number of top document candidates we retain after the residual scoring step and before the full-precision scoring. We test on four datasets, viz., MS-Marco, Fever, HotpotQA and Pooled. Figure 17 summarizes the results. It can be observed that the default DISCo ($n' = 1$) trades off coverage for efficiency. As we increase $n'$, we achieve gains in coverage. In the case of Fever and HotpotQA, the default DISCo configuration provides a better coverage-efficiency curve, while in the case of MS-Marco, $n' = 10$ is the better performing outlier.

## H.6 GREEDY ALGORITHM VARIATIONS

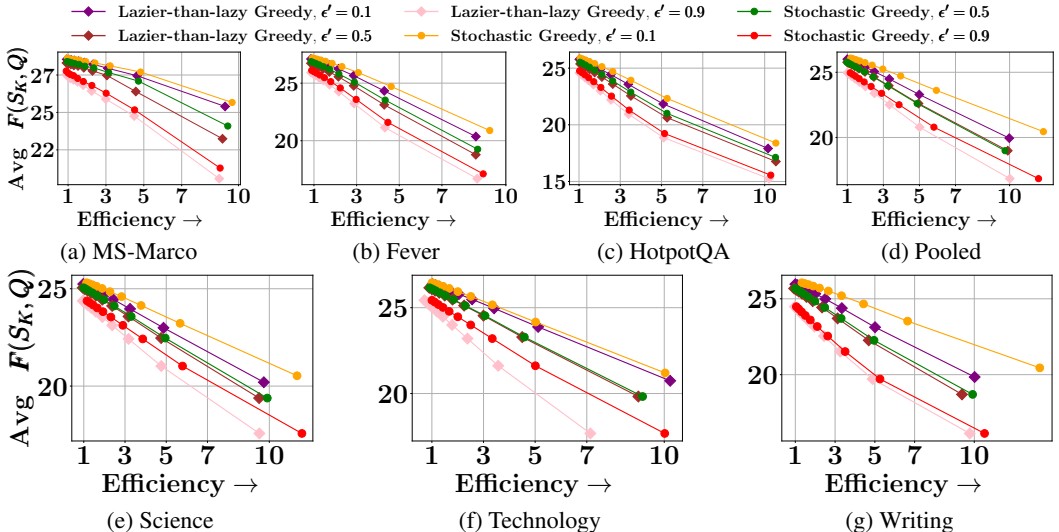

Figure 18: Analysis of coverage-efficiency tradeoffs for Lazier-than-lazy Greedy and Stochastic Greedy for varying $\epsilon'$.

In Figure 18, we provide insights into the performance of Lazier-than-lazy Greedy and Stochastic Greedy across varying $\epsilon'$. We find that Stochastic Greedy achieves the best coverage at $\epsilon' = 0.1$, followed by Lazier-than-lazy Greedy at $\epsilon' = 0.1$. The per-query time taken by each method is dominated by the time to compute the similarity kernel, which results in similar efficiency tradeoffs across all methods.

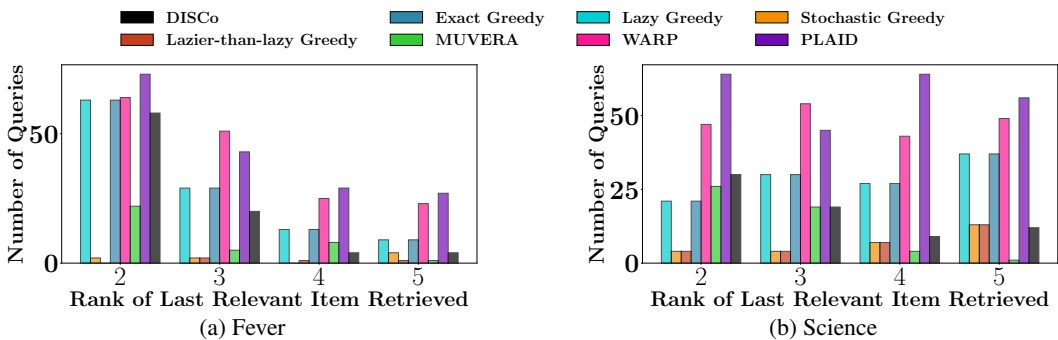

Figure 19: Histogram of the rank of the last ($|S_{\text{gold}}|$-th) true relevant item

### H.7 COVERAGE OBJECTIVE ON GOLD SET OF ITEMS

We extend the experiment on measuring and comparing the coverage objective on the given gold set of items with solutions sets obtained by other methods. Specifically, we provide results on the Fever and Science datasets. We operate on those queries which have at least two relevant documents in the corresponding corpus. The gold set of corpus items for each query is capped at the top two items according to the marginal gain obtained on our coverage function. Figure 19 summarizes the results. We observe that across datasets, PLAID outperforms other methods at gold set retrieval. This is due to the fact that the gold sets on these datasets are meant to be used for independent top-$K$ retrieval. HotpotQA is the exception, and this is because it is a multi-hop dataset amenable to collaborative retrieval.

## I ADDITIONAL EXPERIMENTS DURING REBUTTAL

Here, we provide a consolidated report of the experiments conducted during the rebuttal, as well as any clarifications about our pipeline and the datasets.

### I.1 ADDITIONAL PERFORMANCE MEASURES

In this section, we expand our evaluation in the style of HotpotQA, to two more question-answering style datasets, namely 2WikiMultihopQA (Ho et al., 2020) and Musique (Trivedi et al., 2022). In addition to MAP on the gold labels, we also evaluate the set selection quality using three standard set-overlap metrics. For each query $q$, let $S_q^*$ denote the ground-truth pseudo-relevant set and $S_{q,K}$ the size-$K$ set selected by the method under budget $K$. We report:

**Subset Recall@$K$,** also called *perfect recall at $K$*, the fraction of queries for which the selected set fully contains the ground-truth set: $S_q^* \subseteq S_{q,K}$.

**Precision@$K$,** the average proportion of selected items that are relevant: $\frac{|S_q^* \cap S_{q,K}|}{|S_{q,K}|}$.

**Recall@$K$,** the average proportion of relevant items recovered: $\frac{|S_q^* \cap S_{q,K}|}{|S_q^*|}$.

Together, these metrics capture both strict containment performance (subset recall@K) and standard overlap-based retrieval quality (precision@K and recall@K).

| Dataset | DISCo | Exact Greedy | Lazy Greedy | PLAID | WARP |
|---|---|---|---|---|---|
| 2WikiMultiHopQA | **0.90** | 0.91 | 0.91 | 0.89 | 0.82 |
| Musique | **0.64** | 0.66 | 0.66 | 0.61 | 0.50 |

Table 20: Mean Average Precision (MAP).

| Dataset | **DISCo** | Exact Greedy | Lazy Greedy | PLAID | WARP |
|---|---|---|---|---|---|
| 2WikiMultiHopQA | **0.26** | 0.27 | 0.27 | 0.22 | 0.19 |
| Musique | **0.09** | 0.10 | 0.10 | 0.08 | 0.06 |

Table 21: Subset Recall@K: fraction of queries where $S_q^* \subseteq S_{q,K}$.

| Dataset | **DISCo** | Exact Greedy | Lazy Greedy | PLAID | WARP |
|---|---|---|---|---|---|
| 2WikiMultiHopQA | **0.34** | 0.35 | 0.35 | 0.32 | 0.31 |
| Musique | **0.23** | 0.23 | 0.23 | 0.20 | 0.17 |

Table 22: Precision@K: $\frac{|S_q^* \cap S_{q,K}|}{|S_{q,K}|}$.

| Dataset | **DISCo** | Exact Greedy | Lazy Greedy | PLAID | WARP |
|---|---|---|---|---|---|
| 2WikiMultiHopQA | **0.60** | 0.61 | 0.61 | 0.57 | 0.55 |
| Musique | **0.41** | 0.42 | 0.42 | 0.37 | 0.32 |

Table 23: Recall@K: $\frac{|S_q^* \cap S_{q,K}|}{|S_q^*|}$.

In Tables 20, 21, 22, and 23, we note that DISCo performs closely wrt the greedy baselines and outperforms the IR baselines PLAID and WARP. We also note that on a per-query basis, DISCo takes 153.86 seconds on 2WikiMultihopQA and 42 seconds on Musique, whereas Exact Greedy takes 189.71 seconds on 2WikiMultihopQA and 73 seconds on Musique, leading to speedups of 1.23x and 1.73x respectively.

## I.2 COVERAGE OF GOLD ITEMSETS FOR QA DATASETS

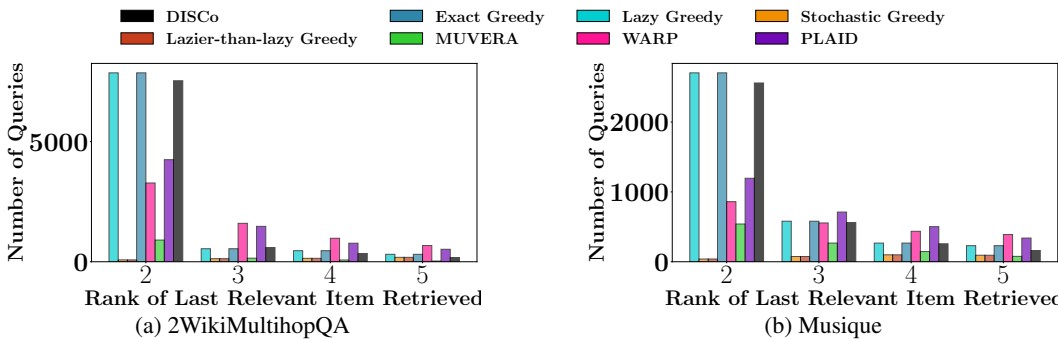

Figure 24: Histogram of the rank of the last ($|S_{\text{gold}}|$-th) true relevant item for the new QA datasets. Left: 2WikiMultihopQA, right: Musique.

We analyze the ranking of the last relevant item across all methods by plotting a histogram of the position of the last gold retrieved item. Note that a low value of the rank is better as then all items in $S_{\text{gold}}$ are retrieved within a small cutoff value of $K$. Figure 24 shows the results: DISCo more frequently retrieves all the gold documents within the top 2–3 positions, performing closely wrt Exact Greedy and Lazy Greedy, whereas IR methods such as MUVERA, PLAID, and WARP more frequently place the last relevant item at lower ranks.

## I.3 PERFORMANCE ON BRIGHT BENCHMARK

Figure 25 shows the trade-off between average coverage objective $\overline{F}_K$ and the efficiency with respect to Exact Greedy. These tradeoffs are obtained by varying the subset size $K$. We note that DISCo achieves the most balanced tradeoff compared the other baselines. DISCo is also vastly more efficient than the greedy baselines (Exact Greedy, Lazy Greedy, etc.). Amongst the IR baselines, PLAID and WARP compete with each other, with PLAID outperforming WARP on the first two datasets, and WARP doing so on the other two.

## I.4 COMPARISON WITH COLBERTV2 AND SPLADE

Figure 26 details the trade-off between average coverage objective $\overline{F}_K$ and the efficiency with respect to Exact Greedy, on a limited subset of the baselines. The key focus is on DISCo's performance in comparison to ColBERTv2 (Santhanam et al., 2021) and SPLADE (Formal et al., 2021). We observe that DISCo provides higher quality tradeoffs compared to both ColBERTv2 and SPLADE. Amongst the other IR baselines, ColBERTv2 and PLAID often perform closely wrt

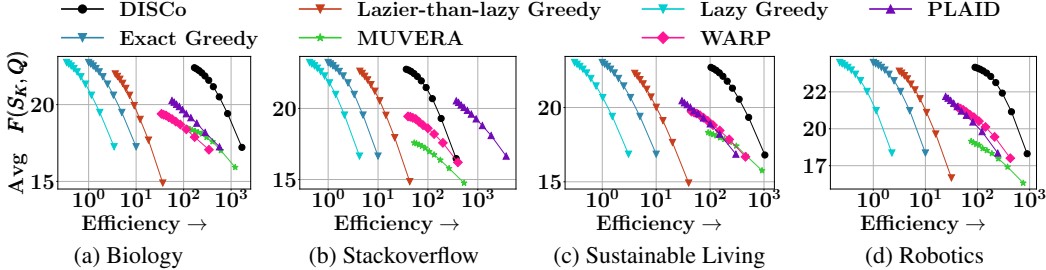

Figure 25: Trade off between efficiency and average coverage objective of DISCO and a subset of state-of-the-art baselines on four datasets: (a) Biology, (b) Stackoverflow, (c) Sustainable Living, and (d) Robotics. All datasets were chosen from the Bright benchmark. DISCO achieves the best trade-off in three datasets, and the best coverage in Stackoverflow against the IR baselines, where the next efficient performer is low on coverage. Efficiency is in log-scale. Upper right corner is the best quadrant.

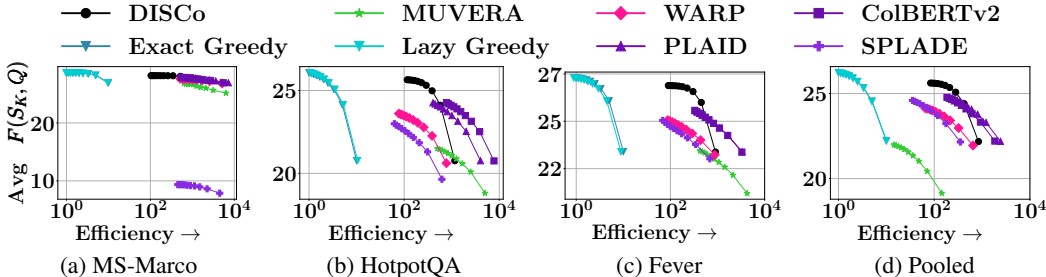

Figure 26: Trade off between efficiency and average coverage objective of DISCO and a subset of the state-of-the-art baselines on four datasets: (a) MS-Marco, (b) HotpotQA, (c) Fever, and (d) Pooled. DISCO achieves the most balanced trade-off, and is able to outperform both new baselines: ColBERTv2 and SPLADE. Upper right corner is the best quadrant.

each other, while WARP and SPLADE are the next best performers. Notably, SPLADE is unable to achieve coverage on MS-Marco, which contains the largest corpus out of the four datasets.

### I.5 FURTHER CLARIFICATION ON THE GOLD SETS OF HOTPOTQA

Here, we reply to the reviewer's concerns about why HotpotQA's gold sets differ from that of the other datasets's.

BEIR (Thakur et al., 2021) collects diverse datasets into a single retrieval benchmark, eliding fine distinctions in the semantics of their relevant item subsets. HotpotQA (Yang et al., 2018) is a multi-hop question answering dataset, which was absorbed into BEIR in order to test zero-shot retrieval. In the original setting, each instance within a split (train/dev/test) contained a query, supporting passages (some of which were relevant and some were "distractors"), a list of candidate answers and the gold answer. Each query required reasoning over all the relevant supporting facts to obtain the answer. An example of such a query is "The mulga apple is often eaten by people who genetic research has inferred a date of habitation as early as when?". This query requires understanding what a mulga apple is, what kinds of people ate them often according to researchers, and when these apple-eaters inhabited the Earth.

When it was repurposed into the BEIR benchmark, the set of queries was kept as-is, while an independent corpus was constructed using Wikipedia. The multi-hop reasoning nature of the dataset remains even in its IR reformation. This distinguishes it from other datasets in the BEIR benchmark, such as FEVER and MSMarco

### I.6 ON HANDLING UPDATES TO THE CORPUS EFFICIENTLY

The topic of efficient corpus updates has practical significance for a wide variety of indexes and vector databases. Updating even classical posting-list–based inverted indexes involves complex engineering, as outlined in Chapter 7 of BCC's text on search engines and information retrieval (Büttcher et al., 2016). Clustered-IVF–style dense indices can be updated for a while by inserting modified

postings into their lists—facing much the same impediments to posting-list compression as lexical inverted indexes—but over time the stale clustering may cause hotspots and require reclustering. Most relevant system-optimization issues are covered in the classic BCC (Büttcher et al., 2016) text. This question encompasses most dense indexes and is not limited to DISCO, and would be an intriguing avenue of future work, but it is far beyond the scope of set retrieval from an unchanging corpus, which itself poses significant challenges that our work addresses.

One way of incorporating new insertions to the corpus without having to perform the entire indexing process from scratch is to perform a virtual insertion during retrieval, i.e., during the replica-level centroid pruning process (Section 3.4). Across each of the $r = 1$ to $R$ replicas of the index, we compute the similarity of the new item $c'$ (using its corresponding embeddings) with the centroids $\{\mathbf{o}\_b, r\}_{b=1}^{B}$ where $B$ is the number of clusters. The item $c'$ is assigned to the cluster corresponding to the highest similarity. Thereafter, pruning and subsequent operations can be executed as usual.

The above solution depends on the volume of inserts being executed within a period of time. For a higher volume of insert operations, more suitable data structures need to be employed. One example of such a data structure is a log structured merge tree (LSM), which writes updates to a primary memory buffer before flushing all of them to an immutable file in secondary memory. Other engineering techniques include batching updates, index versioning and snapshotting.

## I.7    ON THE USE OF MAP IN QA-STYLE DATASETS

We have included a discussion on this matter in Section 4, after "Evaluation setting, coverage and efficiency". We present an elaboration here.

Of all corpus subsets, there is a gold subset $S^*$. Ideally, a set ranking system would return sets $S_1, S_2, \ldots$ in the order of "improving approximation" to $S^*$. In practice, a retrieval system will generally rank items, not subsets. Suppose the items are ranked $x_1, x_2, \ldots$. If we include the first $K$ items in the system output, i.e., $S_K = \{x_1, \ldots, x_K\}$, then there is clear motivation for using MAP: the gold target is a set, and the system outputs a ranked list.

Note that (although its name includes 'precision' and not 'recall') MAP does take recall into account because it averages precision up to the position of the *last* relevant item from $S^*$. For downstream "reasoning" applications such as RAG for multi-passage QA, retrieving the whole of $S^*$ is usually mandatory for correctness, possibly by setting $K$ to be sufficiently large. This renders MRR unacceptable as an evaluation measure, because MRR is content to track the rank of just the *first* ranked item from $S^*$. NDCG does not care about recall of all of $S^*$, either. It is oblivious to loss of recall beyond its top-$K$ horizon.

One might argue that the differential treatment MAP accords to the relevant items is misplaced for downstream set consumers, in which case, we can measure recall and precision at $K$ as usual, averaged over queries. We could also measure the average (over queries) rank at which $S^*$ is completely recalled, or the fraction of queries where we succeed at recalling all of $S^*$ at rank $K$. These are important from the "lost in the middle" perspective.

We feel in our context, where retrieving entire set $S$ is crucial, MAP is more appropriate than both MRR and NDCG. Also of interest is the *last* rank where a relevant item is recalled, because it is a proxy for the (substantial) energy consumed by an LLM when the retrieved subset is presented to its context. setWe also evaluated selection quality using three standard set-overlap metrics as described in Appendix I.1.

## I.8    ON THE PERILS OF PSEUDO-LABEL GENERATION

To evaluate using gold labels on non-QA datasets such as MSMarco and Fever, we prompt Qwen2.5-14B-Instruct LLM to obtain pseudo-labels for these datasets. However, we find that the quality of the labels obtained is not satisfactory, and this leads to a severe degradation in metrics across all methods, not just DISCO. As an example, for the query "Ed Decter only has citizenship in China." from the Fever dataset, the LLM labels a passage about the WCHA's Ice Hockey tournament as a gold item. Ed Decter is an American film director and has no relation to ice hockey. Thus, we avoid the use of pseudo-labels in our work.

### I.9 ON DOWNSTREAM TASK EVALUATION (QUESTION-ANSWERING)

We agree that it would be useful to assess the performance of DISCo on a downstream task such as question answering. To this end, we select two question answering datasets - 2WikiMultihopQA (Ho et al., 2020) (abbreviated 2Wiki) and Musique (Trivedi et al., 2022). We process these datasets for first-stage retrieval, second-stage answering as follows.

For each sample present in the train and dev splits of these datasets, we collect the union of the corresponding supporting facts and generate the corpus. In the case of 2Wiki, this leads to 384,771 unique corpus items and for Musique, 84,453 unique items. Next, we mark the gold corpus items for each query as the ground-truth supporting facts used to answer that query. In the case of 2Wiki, we sample 20K queries u.a.r from the combined train and dev splits, while in the case of Musique, we have 22,355 queries. In the first stage, DISCo retrieves the top-$K$ corpus fact-set for each query according to our coverage score. In the second stage, we prompt the Qwen2.5-14B-Instruct LLM to answer the question given the set of retrieved facts *only* (i.e., without using world knowledge).

We vary $K = 3, 5, 10$ across both datasets, and tabulate our results below.

| Avg Exact Match (EM) | $K = 3$ | $K = 5$ | $K = 10$ |
|---|---|---|---|
| 2Wiki | 0.43 | 0.45 | 0.45 |
| Musique | 0.10 | 0.11 | 0.11 |

Table 27: Average EM scores for 2WikiMultihopQA and Musique.

| Avg F1 Score | $K = 3$ | $K = 5$ | $K = 10$ |
|---|---|---|---|
| 2Wiki | 0.44 | 0.45 | 0.46 |
| Musique | 0.13 | 0.15 | 0.16 |

Table 28: Average F1 scores for 2WikiMultihopQA and Musique.

We find that with increasing $K$, the performance of the LLM improves marginally, indicating the difficulty it faces on dealing with the questions present in both datasets. For example, 2Wiki has four different kinds of questions presenting with varying difficulty (see Table 3 in the 2Wiki paper). On the other hand, all the queries in Musique are of the kind "Which major Russian city borders the body of water in which Saaremaa is located?", and require multiple hops through the corpus to be answered correctly. As a result, while the LLM is able to obtain decent scores on 2Wiki, it is not able to do so on Musique.

Another aspect which adds to the complexity is the fact that the LLM is expected to answer in free-form. No list of candidate answers is given, which renders grounding of the LLM difficult.

### I.10 ON THE CONVERGENCE OF APPROXIMATION OF MARGINAL GAIN AND TRUE GREEDY GAIN

We provide an estimate of the error between the true marginal gain (from greedy) and our approximated marginal gains given number of hyperplanes $R$, across three datasets. In each of the tables, the first column lists which iteration it is we're computing the marginal gain error for. In the remaining columns, we vary $R = 1$ to $5$ and list the gain error. We find that with a modest increase in $R$, the approximated marginal gain and the true greedy gain converge, as desired.

| K→K+1 | R=1 | R=2 | R=3 | R=4 | R=5 |
|---|---|---|---|---|---|
| 1→2 | 0.32 | 0.08 | 0.05 | 0.00 | 0.00 |
| 2→3 | 0.72 | 0.03 | 0.27 | 0.00 | 0.00 |
| 3→4 | 0.75 | 0.04 | 0.10 | 0.00 | 0.00 |
| 4→5 | 0.77 | 0.08 | 0.00 | 0.00 | 0.00 |
| 5→6 | 0.65 | 0.01 | 0.00 | 0.00 | 0.00 |
| 6→7 | 0.47 | 0.00 | 0.00 | 0.00 | 0.00 |
| 7→8 | 0.09 | 0.00 | 0.00 | 0.00 | 0.00 |
| 8→9 | 0.06 | 0.00 | 0.00 | 0.00 | 0.00 |
| 9→10 | 0.05 | 0.00 | 0.00 | 0.00 | 0.00 |

Table 29: Marginal gain error for NFCorpus

| K→K+1 | R=1 | R=2 | R=3 | R=4 | R=5 |
|---|---|---|---|---|---|
| 1→2 | 0.80 | 0.38 | 0.00 | 0.00 | 0.00 |
| 2→3 | 0.31 | 0.31 | 0.31 | 0.31 | 0.02 |
| 3→4 | 0.65 | 0.22 | 0.14 | 0.18 | 0.03 |
| 4→5 | 0.26 | 0.22 | 0.00 | 0.00 | 0.00 |
| 5→6 | 0.02 | 0.03 | 0.00 | 0.00 | 0.00 |
| 6→7 | 0.13 | 0.04 | 0.00 | 0.00 | 0.00 |
| 7→8 | 0.01 | 0.02 | 0.03 | 0.00 | 0.00 |
| 8→9 | 0.14 | 0.03 | 0.00 | 0.00 | 0.00 |
| 9→10 | 0.16 | 0.06 | 0.00 | 0.00 | 0.00 |

Table 30: Marginal gain error for SciFact

| K→K+1 | R=1 | R=2 | R=3 | R=4 | R=5 |
|---|---|---|---|---|---|
| 1→2 | 0.93 | 0.24 | 0.14 | 0.08 | 0.00 |
| 2→3 | 0.70 | 0.00 | 0.00 | 0.00 | 0.00 |
| 3→4 | 0.15 | 0.03 | 0.00 | 0.01 | 0.00 |
| 4→5 | 0.03 | 0.03 | 0.00 | 0.01 | 0.00 |
| 5→6 | 0.02 | 0.02 | 0.01 | 0.02 | 0.00 |
| 6→7 | 0.01 | 0.01 | 0.00 | 0.00 | 0.00 |
| 7→8 | 0.03 | 0.02 | 0.02 | 0.00 | 0.00 |
| 8→9 | 0.03 | 0.02 | 0.01 | 0.01 | 0.00 |
| 9→10 | 0.03 | 0.01 | 0.00 | 0.00 | 0.00 |

Table 31: Marginal gain error for Writing

