# OpenReview forum: "A Dense Subset Index for Collective Query Coverage"
_ICLR.cc/2026/Conference — ICLR 2026 Poster_

### Official Review · Reviewer_8Q8D · 2025-10-29

**Soundness:** 3
**Presentation:** 3
**Contribution:** 3
**Rating:** 6
**Confidence:** 3

**Summary:**

The paper introduces DISCO, a dense retrieval method that selects a subset of documents optimised for query token coverage. Unlike typical top-K retrieval, DISCO selects documents whose embeddings collectively maximise coverage over the query’s token embeddings. The method formalises this as a monotone submodular objective, allowing for a greedy approximation. It then proposes a practical approximation using random projections and a lifted representation, making the approach compatible with ANN search. The authors implement a multi-vector IVF index and show strong empirical performance in terms of coverage and latency on multiple datasets.

**Strengths:**

1. Clear motivation: The paper addresses limitations of independent ranking in multi-hop and multi-evidence tasks.

2. Sound theoretical foundation: The coverage objective is well-defined, submodular, and optimisable via greedy selection.

3. Efficient approximation: The use of lifted representations and random projections to estimate marginal gains is novel and effective.

4. Strong empirical results: On benchmarks like HotpotQA and FEVER, DISCO demonstrates significant latency gains and higher coverage compared to both greedy and top-K baselines.

5. Implementation quality: The multi-stage index is well engineered and extensively ablated.

**Weaknesses:**

1. Lack of downstream evaluation: The paper does not assess end-to-end improvements in downstream tasks such as QA or claim verification. All results focus on coverage and latency.

2. Limited comparative breadth: The baseline comparisons could be broadened to include recent token-aware or set-aware retrieval methods.

3. Unvalidated approximation behaviour: While the greedy coverage algorithm has known guarantees, the practical approximation introduced by DISCO is not empirically analysed against true marginal gains.

**Questions:**

1. Have you tested DISCO in a full downstream setting, such as QA or claim verification, where it serves as the first-stage retriever? It would be useful to know whether the observed coverage improvements actually lead to gains in final task metrics like accuracy or F1.

2. The current baseline set is helpful, but it would be good to understand how DISCO compares to more recent retrieval methods that consider token-level signals or diversity—such as ColBERTv2, GRIP, or SPLADE-style models. Even a brief discussion of these alternatives would help situate the contribution.

3. The approximation strategy for estimating marginal gains is clearly explained, but it would be helpful to see some empirical check on how closely it matches the true greedy gain. For instance, do the approximated and true gains result in similar document rankings or coverage values?

---

> ### Author Response · Authors · 2025-11-21
> **Response to Reviewer 8Q8D (1/3)**
>
> We thank the reviewer for their insightful comments, and provide a point by point rebuttal below.
>
> ## Downstream task evaluation
>
> > Have you tested DISCO in a full downstream setting ... metrics like accuracy or F1.
>
> We agree that it would be useful to assess the performance of DISCo on a downstream task such as question answering. To this end, we select two question answering datasets - 2WikiMultihopQA [A] (abbreviated 2Wiki) and Musique [B]. We process these datasets for first-stage retrieval, second-stage answering as follows.
>
> For each sample present in the train and dev splits of these datasets, we collect the union of the corresponding supporting facts and generate the corpus. In the case of 2Wiki, this leads to 384,771 unique corpus items and for Musique, 84,453 unique items. Next, we mark the gold corpus items for each query as the ground-truth supporting facts used to answer that query. In the case of 2Wiki, we sample 20K queries u.a.r from the combined train and dev splits, while in the case of Musique, we have 22,355 queries. In the first stage, DISCo retrieves the top-$K$ corpus fact-set for each query according to our coverage score. In the second stage, we prompt the Qwen2.5-14B-Instruct LLM to answer the question given the set of retrieved facts _only_ (i.e., without using world knowledge).
>
> We vary $K = 3, 5, 10$ across both datasets, and tabulate our results below.
>
> | Avg Exact Match (EM) | $K = 3$ | $K = 5$ | $K = 10$ |
> | -------- | -------- | -------- | ------- |
> | 2Wiki     |  0.43    |  0.45    |   0.45      |
> | Musique   |  0.10    |  0.11    |   0.11      |
>
> | Avg F1 Score | $K = 3$ | $K = 5$ | $K = 10$ |
> | -------- | -------- | -------- | ------- |
> | 2Wiki     |  0.44    |   0.45   |    0.46     |
> | Musique   |  0.13    |   0.15   |    0.16     |
>
> We find that with increasing $K$, the performance of the LLM improves marginally, indicating the difficulty it faces on dealing with the questions present in both datasets. For example, 2Wiki has four different kinds of questions presenting with varying difficulty (see Table 3 in [A]). On the other hand, all the queries in Musique are of the kind "Which major Russian city borders the body of water in which Saaremaa is located?", and require multiple hops through the corpus to be answered correctly. As a result, while the LLM is able to obtain decent scores on 2Wiki, it is not able to do so on Musique.
>
> Another aspect which adds to the complexity is the fact that the LLM is expected to answer in free-form. No list of candidate answers is given, which renders grounding of the LLM difficult.
>
> [A] Ho, Xanh, et al. "Constructing a multi-hop qa dataset for comprehensive evaluation of reasoning steps." arXiv preprint arXiv:2011.01060 (2020).
>
> [B] Trivedi, Harsh, et al. "Musique: Multihop questions via single-hop question composition, 2022." URL https://arxiv. org/abs/2108.00573 (2021).

---

> ### Author Response · Authors · 2025-11-21
> **Response to Reviewer 8Q8D (2/3)**
>
> ## Limited comparative evaluation
>
> > Limited comparative breadth: The current baseline set is helpful, but ... ColBERTv2, SPLADE ...
>
> We compare DISCo against ColBERTv2, SPLADE, PLAID, WARP on the MSMarco, Fever, HotpotQA and Pooled datasets. We also plot the corresponding coverage-efficiency tradeoff plots; these can be accessed in the Appendix I.4, in Figure 26. The tables below contain the corresponding coverage scores obtained based on a minimum threshold applied to efficiency.
>
> **MSMarco**
> | Method    | Eff ≥ 100 | Eff ≥ 300 | Eff ≥ 450 |
> | --------- | --------- | --------- | --------- |
> | DISCo     | 28.15     | 28.13     | 28.03     |
> | WARP      | 27.62     | 27.62     | 27.62     |
> | PLAID     | 27.85     | 27.85     | 27.85     |
> | ColBERTv2 | 27.87     | 27.87     | 27.87     |
> | SPLADE    | 9.35      | 9.35      | 9.35      |
>
>
> **Fever**
> | Method    | Eff ≥ 100 | Eff ≥ 300 | Eff ≥ 450 |
> | --------- | --------- | --------- | --------- |
> | DISCo     | 26.75     | 26.57     | 26.00     |
> | WARP      | 24.96     | 24.01     | 23.23     |
> | PLAID     | 25.56     | 25.56     | 25.33     |
> | ColBERTv2 | 25.58     | 25.58     | 25.34     |
> | SPLADE    | 24.64     | 23.77     | 23.02     |
>
> **HotpotQA**
> | Method    | Eff ≥ 100 | Eff ≥ 300 | Eff ≥ 450 |
> | --------- | --------- | --------- | --------- |
> | DISCo     | 25.32     | 24.98     | 24.11     |
> | WARP      | 23.43     | 22.26     | 20.61     |
> | PLAID     | 24.26     | 24.26     | 24.10     |
> | ColBERTv2 | 24.27     | 24.27     | 24.27     |
> | SPLADE    | 22.57     | 21.30     | 19.65     |
>
> **Pooled**
> | Method    | Eff ≥ 100 | Eff ≥ 150 | Eff ≥ 250 |
> | --------- | --------- | --------- | --------- |
> | DISCo     | 25.35     | 25.35     | 25.08     |
> | WARP      | 23.93     | 23.65     | 22.97     |
> | PLAID     | 24.74     | 24.74     | 24.69     |
> | ColBERTv2 | 24.78     | 24.78     | 24.56     |
> | SPLADE    | 23.71     | 23.23     | 22.15     |
>
> We note that DISCo achieves the highest coverage given the efficiency cutoffs across all datasets. On certain datasets, a noticeable drop in coverage occurs for the baselines, but DISCo's coverage remains steady. For e.g., on the HotpotQA dataset, WARP drops significantly with an increase in the efficiency threshold compared to DISCo.
>
> Additionally, DISCo remains highly efficient even under strict minimum-coverage constraints. For example, on the MSMarco, Fever, HotpotQA, and Pooled datasets, DISCo achieves speedups of 500x, 300x, 290x, and 280x, respectively, at coverage thresholds of 28, 26, 25, and 25. In contrast, all competing baselines fail to reach these coverage levels at all.

---

> ### Author Response · Authors · 2025-11-21
> **Response to Reviewer 8Q8D (3/3)**
>
> ## Unvalidated approximation behaviour
>
> > The approximation strategy for estimating marginal gains ... how closely it matches true greedy gain ...
>
> We provide an estimate of the error between the true marginal gain (from greedy) and our approximated marginal gains given number of hyperplanes $R$, across three datasets. In each of the tables, the first column lists which iteration it is we're computing the marginal gain error for. In the remaining columns, we vary $R = 1$ to $5$ and list the gain error. We find that with a modest increase in $R$, the approximated marginal gain and the true greedy gain converge, as desired.
>
>
> **NFCorpus**
> | K→K+1 | R=1  | R=2 | R=3 | R=4 | R=5 |
> |-------|------|-----|-----|-----|-----|
> | 1→2   | 0.32 | 0.08 | 0.05 | 0.00 | 0.00 |
> | 2→3   | 0.72 | 0.03 | 0.27 | 0.00 | 0.00 |
> | 3→4   | 0.75 | 0.04 | 0.10 | 0.00 | 0.00 |
> | 4→5   | 0.77 | 0.08 | 0.00 | 0.00 | 0.00 |
> | 5→6   | 0.65 | 0.01 | 0.00 | 0.00 | 0.00 |
> | 6→7   | 0.47 | 0.00 | 0.00 | 0.00 | 0.00 |
> | 7→8   | 0.09 | 0.00 | 0.00 | 0.00 | 0.00 |
> | 8→9   | 0.06 | 0.00 | 0.00 | 0.00 | 0.00 |
> | 9→10  | 0.05 | 0.00 | 0.00 | 0.00 | 0.00 |
>
>
> **SciFact**
> | K→K+1 | R=1  | R=2 | R=3 | R=4 | R=5 |
> |-------|------|-----|-----|-----|-----|
> | 1→2   | 0.80 | 0.38 | 0.00 | 0.00 | 0.00 |
> | 2→3   | 0.31 | 0.31 | 0.31 | 0.31 | 0.02 |
> | 3→4   | 0.65 | 0.22 | 0.14 | 0.18 | 0.03 |
> | 4→5   | 0.26 | 0.22 | 0.00 | 0.00 | 0.00 |
> | 5→6   | 0.02 | 0.03 | 0.00 | 0.00 | 0.00 |
> | 6→7   | 0.13 | 0.04 | 0.00 | 0.00 | 0.00 |
> | 7→8   | 0.10 | 0.02 | 0.03 | 0.03 | 0.00 |
> | 8→9   | 0.14 | 0.03 | 0.00 | 0.00 | 0.00 |
> | 9→10  | 0.16 | 0.06 | 0.00 | 0.00 | 0.00 |
>
> **Writing**
> | K→K+1 | R=1  | R=2 | R=3 | R=4 | R=5 |
> |-------|------|-----|-----|-----|-----|
> | 1→2   | 0.93 | 0.24 | 0.14 | 0.08 | 0.00 |
> | 2→3   | 0.70 | 0.00 | 0.00 | 0.00 | 0.00 |
> | 3→4   | 0.15 | 0.03 | 0.00 | 0.00 | 0.00 |
> | 4→5   | 0.03 | 0.03 | 0.00 | 0.01 | 0.00 |
> | 5→6   | 0.02 | 0.02 | 0.01 | 0.02 | 0.00 |
> | 6→7   | 0.01 | 0.01 | 0.00 | 0.00 | 0.00 |
> | 7→8   | 0.03 | 0.02 | 0.02 | 0.00 | 0.00 |
> | 8→9   | 0.03 | 0.02 | 0.02 | 0.01 | 0.00 |
> | 9→10  | 0.03 | 0.01 | 0.00 | 0.00 | 0.00 |

---

> > ### Author Response · Authors · 2025-11-27
> >
> > Dear Reviewer 8Q8D,
> >
> >
> > We would like to thank you for your review. We would be grateful if you could take look into our response and let us know if your concerns are addressed. If you need any further clarification, please feel free to let us know.
> >
> >
> > Regards,
> >
> > Authors

---

### Official Review · Reviewer_qUAz · 2025-10-31

**Soundness:** 3
**Presentation:** 3
**Contribution:** 2
**Rating:** 4
**Confidence:** 4

**Summary:**

The paper correctly recognizes that many ranking algorithms rank items independently of one another, so that important aspects of query coverage may be missed if only the top K items are retrieved. This is absolutely true and further study on this problem is definitely worthwhile, especially from the perspective of modern dense retrieval.

Unfortunately, this work appears to have been conducted in a bit of a vacuum, without any clear consideration of the huge volume of prior work in this area, especially related work on learning to rank for web search. The section on monotonicity, submodularity, and greedy maximization could have be lifted without change from a paper appearing 10 or more years ago. I included a bunch of examples below.

I also have concerns about the evaluation. For coverage, only HotpotQA has gold labels, which limits the evaluation. However, there are various methods for employing LLMs to generate pseudo-labels. While it's not exactly what you want Ragnarok (https://arxiv.org/abs/2406.16828) can probably be adapted to extract nuggets for coverage.

MAP is very strange measure to use on these collection. NDCG and MRR are more standard.

Rodrygo L.T. Santos, Craig Macdonald, and Iadh Ounis. 2010. Exploiting query reformulations for web search result diversification. In Proceedings of the 19th international conference on World wide web (WWW '10). Association for Computing Machinery, New York, NY, USA, 881–890. https://doi.org/10.1145/1772690.1772780

Cheng Xiang Zhai, William W. Cohen, and John Lafferty. 2003. Beyond independent relevance: methods and evaluation metrics for subtopic retrieval. In Proceedings of the 26th annual international ACM SIGIR conference on Research and development in informaion retrieval (SIGIR '03). Association for Computing Machinery, New York, NY, USA, 10–17. https://doi.org/10.1145/860435.860440

Jun Xu, Long Xia, Yanyan Lan, Jiafeng Guo, and Xueqi Cheng. 2017. Directly Optimize Diversity Evaluation Measures: A New Approach to Search Result Diversification. ACM Trans. Intell. Syst. Technol. 8, 3, Article 41 (May 2017), 26 pages. https://doi.org/10.1145/2983921

Learning for Search Result Diversification. Yadong Zhu Yanyan Lan Jiafeng Guo Xueqi Cheng Shuzi Niu

These are just random papers that popped to mind. I think there was some good work by Olivier Chapelle but all I can find is this workshop paper: Paul N. Bennett, Ben Carterette, Olivier Chapelle, and Thorsten Joachims. 2008. Beyond binary relevance: preferences, diversity, and set-level judgments. SIGIR Forum 42, 2 (December 2008), 53–58. https://doi.org/10.1145/1480506.1480516

*Note that the last one is 2008*

**Strengths:**

The topic is important, and the theory seems correct. I think there are novel aspects, and certainly a version of this paper should be published at some point.

The focus on efficiency is good to see.

**Weaknesses:**

What I said in the summary: There's a huge thread of similar work that has been ignored. The evaluation is not appropriate to the problem.

**Questions:**

Can you clarify the connection to past work?

Can you defend the limitations in the evaluation raised in the summary?

---

> ### Author Response · Authors · 2025-11-21
> **Response to Reviewer qUAz (1/3)**
>
> We thank the reviewer for their careful review. We address their concerns below.
>
> ## Related work
>
> > huge volume of prior work … on learning to rank for web search … monotonicity, submodularity, and greedy maximization
>
> Thanks for calling this out.
>
> We point out that the para about monotonicity, submodularity, and greedy maximization (second paragraph of Section 3.1) are acknowledged to be standard. We included this material to make the paper self-complete and easily understood by reviewers with diverse backgrounds.
>
> We rewrote the related work section (3.2 now) with a better perspective of pre-existing work and the exact scope of our innovation.  For convenience we include an excerpt here.
>
> Submodular set reward functions have been proposed in the Information Retrieval community since at least 1998, motivated by diversity [Bennett, 2008] and subtopic coverage [Zhai, 2003]. These objectives are usually implemented as a reranking stage, after the small subset of candidates has already been selected using a scalable first-stage retriever. For reranking, max marginal relevance [Carbonell, 1998], multi-armed bandits [Radlinski, 2008], determinantal point processes [Kulesza, 2012; Chen, 2017], query reformulation [Santos, 2010], etc., are used. Hence, these approaches focus on scoring function computation at the reranking stage. These reranking efforts are vulnerable to loss of recall in the first-stage.
>
> In contrast, our focus is on coverage in the first stage itself, where we design indexing and retrieval method tailored specifically for coverage maximization. Note that, submodular maximization has been widely used since 1978, but our work focus on designing ANN retriever for coverage based submodular maximization. Therefore, our work focus on indexing and search, whereas these existing works, albeit related, focus on suitable submodular scoring function computation and the application of greedy variants to maximize it.  Extensive search reveals a paucity on direct _first-stage dense retrievers that optimize a query-coverage objective_. A notable exception is in the use of pseudo-relevance feedback in dense retrieval to improve facet/subtopic coverage [Yu, 2021] ― like key-value memory networks [Miller, 2016], they also perform multi-round dense query modification, but there are no formal coverage guarantees.

---

> ### Author Response · Authors · 2025-11-21
> **Response to Reviewer qUAz (2/3)**
>
> ## Evaluation methodology
>
> > MAP is very strange measure to use on these collection. NDCG and MRR are more standard.
>
> We have included a discussion on this matter in Section 4, after "Evaluation setting, coverage and efficiency" in the updated manuscript.  We present an elaboration here.
>
> Of all corpus subsets, there is a gold subset $S^\*$. Ideally, a set ranking system would return sets $S_1, S_2, \ldots$ in the order of “improving approximation” to $S^\*$. In practice, a retrieval system will generally rank items, not subsets. Suppose the items are ranked $x_1, x_2, \ldots$. If we include the first $K$ items in the system output, i.e., $S_K = \{x_1, \ldots, x_K\}$, then there is clear motivation for using MAP: the gold target is a set, and the system outputs a ranked list.
>
> Note that (although its name includes ‘precision’ and not ‘recall’) MAP does take recall into account because it averages precision up to the position of the _last_ relevant item from $S^\*$. For downstream “reasoning” applications such as RAG for multi-passage QA, retrieving the whole of $S^\*$ is usually mandatory for correctness, possibly by setting $K$ to be sufficiently large. This renders MRR unacceptable as an evaluation measure, because MRR is content to track the rank of just the _first_ ranked item from $S^\*$. NDCG does not care about recall of all of $S^\*$, either. It is oblivious to loss of recall beyond its top-$K$ horizon.
>
> One might argue that the differential treatment MAP accords to the relevant items is misplaced for downstream set consumers, in which case, we can measure recall and precision at $K$ as usual, averaged over queries. We could also measure the average (over queries) rank at which $S^\*$ is completely recalled, or the fraction of queries where we succeed at recalling all of $S^\*$ at rank $K$. These are important from the “lost in the middle” perspective.
>
> We feel in our context, where retrieving entire set $S$ is crucial, MAP is more appropriate than both MRR and NDCG. Also of interest is the _last_ rank where a relevant item is recalled, because it is a proxy for the (substantial) energy consumed by an LLM when the retrieved subset is presented to its context.
>
> During the rebuttal period, we also evaluated selection quality using three standard set-overlap metrics. For each query $q$, let $S_q^\*$ denote the ground-truth pseudo-relevant set and $S_{q,K}$ the size-$K$ set selected by the method under budget $K$. We report:
>  * $\textbf{Subset Recall@K}$, the fraction of queries for which the selected set fully contains the ground-truth set: $S_q^\* \subseteq S_{q,K}$.
>  * $\textbf{Precision@K}$, the average proportion of selected items that are relevant: $\frac{|S_q^\* \cap S_{q,K}|}{|S_{q,K}|}$.
>  * $\textbf{Recall@K}$, the average proportion of relevant items recovered: $\frac{|S_q^\* \cap S_{q,K}|}{|S_q^\*|}$.
>
> Together, these metrics capture both strict containment performance (subset recall@K) and standard overlap-based retrieval quality (precision@K and recall@K).

---

> ### Author Response · Authors · 2025-11-21
> **Response to Reviewer qUAz (3/3)**
>
> ## Pseudo-label generation
>
> > ... concerns about the evaluation ... employing LLMs to generate pseudo-labels ...
>
> We agree with the reviewer on the need for further evaluation with gold labels. To this end, we prompt Qwen2.5-14B-Instruct LLM to obtain pseudo-labels for MSMarco and Fever datasets. However, we find that the quality of the labels obtained is not satisfactory, and this leads to a severe degradation in metrics across all methods, not just DISCo.
>
> As an example, for the query "Ed Decter only has citizenship in China." from the Fever dataset, the LLM labels a passage about the WCHA's Ice Hockey tournament as a gold item. Ed Decter is an American film director and has no relation to ice hockey.
>
> ## Additional results
>
> To work around this problem, we perform our evaluations on two more multi-hop datasets in the same vein as HotpotQA, namely 2WikiMultihopQA [A] (abbreviated 2Wiki) and Musique [B]. We compute the MAP on the obtained ranked list, and we use rank as the selection order for greedy methods and MaxSim ranks for the remaining IR baselines. Further, we compute three new metrics described above. We present additional results in the newly-added Appendix I in the PDF manuscript. Here we provide some excerpts in tabular form.
>
> **MAP**
> | Dataset         | DISCo | Exact Greedy | Lazy Greedy | PLAID | WARP |
> | --------------- | ----- | ------------ | ----------- | ----- | ---- |
> | 2WikiMultiHopQA | 0.90  | 0.91         | 0.91        | 0.89  | 0.82 |
> | Musique         | 0.64  | 0.66         | 0.66        | 0.61  | 0.50 |
>
>
> **Subset Recall@K**
> | Dataset         | DISCo | Exact Greedy | Lazy Greedy | PLAID | WARP |
> | --------------- | ----- | ------------ | ----------- | ----- | ---- |
> | 2WikiMultiHopQA | 0.26  | 0.27         | 0.27        | 0.22  | 0.19 |
> | Musique         | 0.09  | 0.10         | 0.10        | 0.08  | 0.06 |
>
>
> **Precision@K**
> | Dataset         | DISCo | Exact Greedy | Lazy Greedy | PLAID | WARP |
> | --------------- | ----- | ------------ | ----------- | ----- | ---- |
> | 2WikiMultiHopQA | 0.34  | 0.35         | 0.35        | 0.32  | 0.31 |
> | Musique         | 0.23  | 0.23         | 0.23        | 0.20  | 0.17 |
>
>
> **Recall@K**
> | Dataset         | DISCo | Exact Greedy | Lazy Greedy | PLAID | WARP |
> | --------------- | ----- | ------------ | ----------- | ----- | ---- |
> | 2WikiMultiHopQA | 0.60  | 0.61         | 0.61        | 0.57  | 0.55 |
> | Musique         | 0.41  | 0.42         | 0.42        | 0.37  | 0.32 |
>
> Across all datasets and metrics, we find that DISCo performs closely wrt the greedy baselines and outperforms the IR baselines PLAID and WARP. We also note that on a per-query basis, DISCo takes 153.86 seconds on 2Wiki and 42 seconds on Musique, whereas Exact Greedy takes 189.71 seconds on 2Wiki and 73 seconds on Musique, leading to speedups of 1.23x and 1.73x respectively.
>
> [A] Ho, Xanh, et al. "Constructing a multi-hop qa dataset for comprehensive evaluation of reasoning steps." arXiv preprint arXiv:2011.01060 (2020).
>
> [B] Trivedi, Harsh, et al. "Musique: Multihop questions via single-hop question composition, 2022." URL https://arxiv. org/abs/2108.00573 (2021).

---

> > ### Comment · Reviewer_qUAz · 2025-11-22
> >
> > I appreciate the effort the authors have made to address my concerns. If the paper is revised as indicated, I support acceptance.

---

> > > ### Author Response · Authors · 2025-11-28
> > >
> > > Thanks for your encouraging comments and increasing the score. We have already included all the discussions in the paper.

---

### Official Review · Reviewer_htMi · 2025-11-01

**Soundness:** 3
**Presentation:** 3
**Contribution:** 3
**Rating:** 8
**Confidence:** 4

**Summary:**

This paper proposes a novel reformulation of the classical task of information retrieval. While traditional IR is optimized for selecting a single "best" document out of a corpus of choices, the authors note that these techniques are ill-suited for modern, reasoning intensive retrieval tasks such as multi-hop question answering.

To address this challenge, the authors propose to reframe retrieval as a set coverage problem and cast the task of finding the item that maximizes the marginal gain of query coverage as a multi-vector retrieval task. The authors implement their proposed pipeline in a system called DISCo which achieves significantly improved query-latency tradeoff metrics over a host of established retrieval baselines.

**Strengths:**

The paper presents a novel retrieval algorithm that achieves improved performance on reasoning-intensive retrieval tasks by creatively combining submodular optimization with multi-vector retrieval. While none of the components of the authors' proposed architecture is entirely novel, the combination of these techniques is original and appears to achieve state-of-the-art performance across a number of standard retrieval baselines in this area. The authors' proposed technique is also principled and they also provide a number of theoretical guarantees. The paper is also very well-written and motivates the problem well.

**Weaknesses:**

1. While the baselines considered in the experiments is quite thorough, the claims in the paper might be better supported by considering additional retrieval benchmarks specifically tailored towards challenging retrieval-intensive tasks, such as the recently introduced [Bright Benchmark](https://arxiv.org/pdf/2407.12883). Evaluating on benchmarks specifically designed for reasoning-intensive tasks (as opposed to more generic retrieval like BEIR) might be a more natural fit for this paper and might help us better understanding the strengths and limitations of the DISCo method.

2. The paper is well-written but I found it a bit hard to understand how all the pieces fit together on a first read through the paper. The authors might want to consider adding a schematic diagram or conceptual figure to aid in communicating their methodology.

3. I think the main body of the paper could benefit from a dedicated section discussing the relevant related work (especially in introducing the baseline methods evaluated later in the experimental section).

4. The authors could also make their theoretical contributions more clear by perhaps introducing informal statements of the theorems they prove instead of relegating all of this content to the appendix (which readers may not read and thus never find).

**Questions:**

1. Can you provide more clarification on the "gold set" of hotpotqa? Why is this different from the other benchmark datasets used in the experiments section?

2. Can you consider evaluating DISCo on the [Bright Benchmark](https://arxiv.org/pdf/2407.12883) and possibly other related reasoning-intensive retrieval benchmarks?

3. Do existing methods handle updates to the corpus efficiently? How would you assess this limitation of your method in relation to the existing literature?

---

> ### Author Response · Authors · 2025-11-21
> **Response to Reviewer htMi (1/2)**
>
> We thank the reviewer for their positive review. We address their remaining concerns below.
>
> ## Evaluation using the Bright benchmark
>
> We compare DISCo against three greedy baselines and PLAID, MUVERA and WARP, on four datasets from the Bright benchmark, namely Biology, Stackoverflow, Sustainable Living and Robotics. We plot the related coverage-efficiency plots, which are available in the Appendix I.3. The tables below contain the corresponding coverage scores obtained based on a minimum threshold applied to efficiency. Here, efficiency is measured as ratio of the time taken by greedy algorithm and the corresponding method.
>
> **Biology**
> | Method | Eff ≥ 100 | Eff ≥ 300 | Eff ≥ 450 |
> | ------ | --------- | --------- | --------- |
> | DISCo  | 22.40     | 21.58     | 20.50     |
> | WARP   | 18.39     | 17.06     | –         |
> | PLAID  | 19.40     | 17.25     | 17.25     |
> | MUVERA | 18.46     | 17.87     | 17.01     |
>
>
> **Stackoverflow**
> | Method | Eff ≥ 50 | Eff ≥ 75 | Eff ≥ 100 |
> | ------ | -------- | -------- | --------- |
> | DISCo  | 22.46    | 21.47    | 20.69     |
> | WARP   | 19.26    | 18.83    | 18.20     |
> | PLAID  | 20.54    | 20.54    | 20.54     |
> | MUVERA | 17.58    | 17.35    | 16.97     |
>
> **Sustainable Living**
> | Method | Eff ≥ 100 | Eff ≥ 200 | Eff ≥ 350 |
> | ------ | --------- | --------- | --------- |
> | DISCo  | 22.70     | 21.77     | 19.34     |
> | WARP   | 18.81     | 17.85     | 16.68     |
> | PLAID  | 18.90     | 16.86     | –         |
> | MUVERA | 18.25     | 17.68     | 16.79     |
>
>
> **Robotics**
> | Method | Eff ≥ 100 | Eff ≥ 150 | Eff ≥ 300 |
> | ------ | --------- | --------- | --------- |
> | DISCo  | 24.03     | 23.54     | 21.15     |
> | WARP   | 20.52     | 19.45     | 18.01     |
> | PLAID  | 19.80     | 18.35     | –         |
> | MUVERA | 18.89     | 18.53     | 17.42     |
>
> We note that DISCo achieves the best coverage scores given the efficiency threshold across all datasets. In certain cases, the baselines are unable to achieve any coverage for the given efficiency cutoff, for e.g. PLAID in Robotics.  Results in Appendix I.3 also suggests that DISCo is often efficient even when a high coverage is desired. For example, for a coverage cutoff of 20, DISCo achives 500x efficiency, while the other baselines are unable to achieve that coverage.
>
> ## System diagram, related work
>
> > Need for a schematic diagram, Related Work section and theoretical contributions in the main body of the paper
>
> Thanks for this suggestion. We have provided a schematic diagram of the DISCo system in Figure 1, page 6 of the updated manuscript. We also provide a new Related Work section (3.2) in the main paper, and proof sketches for theorems 2 and 3.

---

> ### Author Response · Authors · 2025-11-21
> **Response to Reviewer htMi (2/2)**
>
> ## "Gold set" $S^*$ clarification
>
> > Clarification on the "gold set" of HotpotQA
>
> BEIR collects diverse datasets into a single retrieval benchmark, eliding fine distinctions in the semantics of their relevant item subsets. HotpotQA is a multi-hop question answering dataset, which was absorbed into BEIR in order to test zero-shot retrieval. In the original setting, each instance within a split (train/dev/test) contained a query, supporting passages (some of which were relevant and some were "distractors"), a list of candidate answers and the gold answer. Each query required reasoning over _all_ the relevant supporting facts to obtain the answer. An example of such a query is "The mulga apple is often eaten by people who genetic research has inferred a date of habitation as early as when?". This query requires understanding what a mulga apple is, what kinds of people ate them often according to researchers, and when these apple-eaters inhabited the Earth.
>
> When it was repurposed into the BEIR benchmark, the set of queries was kept as-is, while an independent corpus was constructed using Wikipedia. The multi-hop reasoning nature of the dataset remains even in its IR reformation. This separates it from other datasets in the BEIR benchmark, such as FEVER and MSMarco.
>
> While the tradeoff between coverage and efficiency is our primary target for evaluation (Section 4.1, "Trade-off between coverage and efficiency"), data sets like HotpotQA compel us to perform additional evaluation of direct recall of $S^*$ in such cases (Section 4.1, "Does $F(S, Q)$ reward $S_{gold}$?").
>
>
> ## Handling corpus updates
>
> > Efficiently handling updates to the corpus
>
> This question is of great practical significance and applies to many dense indexes or vector DBs. Updating even classical posting-list–based inverted indexes involves complex engineering, as outlined in Chapter 7 of [Bütcher, Clarke, and Cormack]. Clustered-IVF–style dense indices can be updated for a while by inserting modified postings into their lists—facing much the same impediments to posting-list compression as lexical inverted indexes—but over time the stale clustering may cause hotspots and require reclustering. Most relevant system-optimization issues are covered in the classic [Bütcher, Clarke, and Cormack] text. This question encompasses most dense indexes and is not limited to DISCo, and would be an intriguing avenue of future work, but it is far beyond the scope of set retrieval from an unchanging corpus, which itself poses significant challenges that our work addresses.
>
> https://plg.uwaterloo.ca/~ir/ir/book/
>
> One way of incorporating new insertions to the corpus without having to perform the entire indexing process from scratch is to perform a virtual insertion during retrieval, i.e., during the replica-level centroid pruning process (ref Section 3.4 of the paper). Across each of the $r = 1$ to $R$ replicas of the index, we compute the similarity of the new item $c'$ (using its corresponding embeddings) with the centroids $\\{\mathbf{o}\_{b,r}\\}_{b=1}^B$ where $B$ is the number of clusters. The item $c'$ is assigned to the cluster corresponding to the highest similarity. Thereafter, pruning and subsequent operations can be executed as usual.
>
> The above solution depends on the volume of inserts being executed within a period of time. For a higher volume of insert operations, more suitable data structures need to be employed. One example of such a data structure is a log structured merge tree (LSM), which writes updates to a primary memory buffer before flushing all of them to an immutable file in secondary memory. Other engineering techniques include batching updates, index versioning and snapshotting.

---

### Author Response · Authors · 2025-12-03
**Address to AC**

Dear AC,


Modern multi-hop QA and text-to-SQL require collaborative retrieval: multiple items must jointly cover the query’s contextual vectors. DISCo formulates this as a submodular coverage problem, which a greedy algorithm can approximately maximize. We cast each greedy iteration, which selects an item maximizing marginal gain, as a (dense) retrieval problem. Here we approximate the marginal gain as a dot product of judiciously lifted query and corpus word vectors, followed by a novel random projection. This allows us to design an indexing and search method, leading to a super efficient (often 100x faster than several baselines) coverage based retrieval method.

Reviewers were mostly positive (initial scores: htMi=8, qUAz=4, 8Q8D=6) pre-rebuttal. During rebuttal we addressed each and every concern. The key points we addressed include:

* Evaluation on four datasets from the Bright benchmark, which shows that our method provide the best trade off in terms of coverage and efficiency [htMi]

* Significant elaboration of related work with respect to greedy maximization for submodular functions in the context of learning to rank for web search [qUAz].

* Clarification on why Mean Average Precision (MAP) is more appropriate for evaluation, compared to NDCG and MRR [qUAz].

* Evaluation on two more question answering (QA) based datasets (2WikiMultihopQA and Musique) using a new suite of non-MAP metrics, demonstrating utility of our method [qUAz]

*  Application on downstream QA, comparison with ColBERTv2 and SPLADE on several datasets (which showed our method performs best) [8Q8D]

For more and other details, please refer to the discussions.

---

After the rebuttal, qUAz increased their score from 4 to 8, resulting in scores **8-8-6**.
Unfortunately, the recent reversal brought it back to 8,4,6 again.

 We stressed on the scores because

(1) average score of 7.33 would place a paper well above the poster bracket in normal circumstances.

(2) our rebuttal comprehensively addressed reviewer qUAz's concerns, leading them **to raise their score from 4 to 8** which reflects a substantial upgrade in their opinion of the work. Reviewer quAz has written that they raised their support to **acceptance**  (in contrast to Borderline reject as is reflected by the pre-rebuttal score of 4).

We kindly request that you ensure this post-rebuttal consensus is considered during the decision process.

While this response is now addressed to AC, we are keeping it visible to all reviewers for full transparency. Note that we took explicit permission from the PC chairs to mention the post-rebuttal scores.






Regards,

Author

---

### Meta-Review · Area_Chair_EPh2 · 2026-01-07

**Summary:**

All reviews for this paper are positive except one, which became positive after the rebuttal response. Thus, the paper is a clear accept. The reviewers appreciated the novelty of the approach, the good empirical evaluation, and the good presentation of the paper.

**Reviewer Concerns:**

A major concern of one reviewer was a lack of discussion of prior work on information retrieval that uses similar approaches based on submodularity and greedy selection for coverage. The authors clarify the connection of their work to the prior work -- a key difference is that they target coverage at the initial retrieval stage, not just at reranking, and thus must contribute an efficient index for retrieving documents according to the greedy according to the greedy approach. This is a major novel contribution of the paper.

Other concerns related to the empirical evaluation -- suggesting additional benchmarks to consider, additional baselines to add, and experiments on downstream tasks. These were mostly addressed during the rebuttal, with the notable exception of performance on downstream tasks, for which only very preliminary results were shown.

**Reviewer Scores:**

Reviewer qUAz indicated increasing their score from a 4 to supporting acceptance. I have taken this into account in the decision.

The other reviewers were positive to begin with, and likely would not have increased their scores.

---

### Decision · Program_Chairs · 2026-01-26

Accept (Poster)